# Variant U1 snRNAs contribute to cell cycle and differentiation control of human iPS cells

Yajie Zhu[1,6], Konstantinos Sofiadis[1,2,4,6], Athanasia Mizi [1], Vasilisa Kalinkina [1], Matthias Akyel[1], Milos Nikolic[2,5], Lukas Cyganek [3], Carmelo Ferrai [1] & Argyris Papantonis [1,2] ✉

The maintenance of stem cell identity, as well as the differentiation of stem cells into any lineage, requires precise regulation of gene expression. Despite intensive research, our understanding of these regulatory processes remains incomplete. Here, we focus on the understudied paralogs of the U1 small nuclear RNA gene known as variant U1 snRNAs. By generating isogenic knockout lines of human induced pluripotent stem cells for different variant U1s, we show that their loss profoundly changes both gene expression and cell cycle profiles. These effects manifest alongside alternative splicing patterns, including those involving recursive splicing sites, and lead to differential availability of stem cell regulators. Together, our results shed new light on the functional roles of variant U1 snRNAs and further our understanding of the programs controlling human pluripotency.

The maintenance of stem cell identity and the commitment of stem cells to a differentiation path both rely on the activation of specific gene regulatory networks[1]. Shifting from one to the other involves, amongst others, changes in cell cycle control, cell adhesion and morphology, metabolism, and protein availability[2–6]. Such changes can be achieved by the differential control of gene expression, as well as by changes to alternative splicing patterns. However, the molecular events and the key players leading to these changes are still not fully understood.

Transcription factors, epigenetic regulators and the ensuing control of transcription have been intensely studied with respect to stem cell maintenance and differentiation[7–9], while alternative splicing and its regulation did not receive attention until much more recently[10,11]. Splicing is executed by ribonucleoprotein (RNP) complexes containing small nuclear RNAs (snRNAs) like U1, U2, U4-6 and associated factors, the precise composition and dynamic functional states of which are well described[12,13]. However, various animal genomes were found to carry additional snRNA gene copies that were long considered pseudogenes[14–17]. Strikingly, most of these variant snRNA genes show the highest expression during the earlier stages of development[18], but their functions remain enigmatic.

In humans, almost 1700 copies of snRNA gene variants have been identified to date[19]. They correspond to essentially all major (i.e., U1, U2, U4–6) and minor spliceosome snRNA-encoding genes (i.e., U11, U12, U4[ATAC], U6[ATAC]) and accumulating evidence argues for their incorporation into active spliceosomal complexes and involvement in RNA processing[20–24]. For variant U1 (vU1) genes, >100 annotated copies exist on chr1[22,25]. When compared to the canonical U1 snRNA, various sequence differences, including base substitutions and deletions, arise, but only a few overlap the Sm binding motif[19]. This can potentially alter their binding specificity on pre-mRNAs while allowing for the formation of snRNP complexes with known spliceosome components[24,26].

Human vU1 snRNAs were shown to be highly expressed in pluripotent or reprogrammed stem cells[19], but their roles with respect to stem cell identity and functions remain unknown. To address this, we generated knockout human induced pluripotent stem cell (hiPSC) lines for different vU1 genes and used genomics and functional assays to investigate their effects on hiPSC homeostasis. This way, we were able to show that vU1 loss-of-function triggers marked gene expression and cell cycle changes that can be linked to perturbed alternative splicing (AS) patterns genome-wide.

[1]Institute of Pathology, University Medical Center Göttingen, Göttingen, Germany. [2]Center for Molecular Medicine Cologne, University of Cologne, Cologne, Germany. [3]Stem Cell Unit, Clinic for Cardiology and Pneumology, University Medical Center Göttingen, Göttingen, Germany. [4]Present address: Princess Maxima Center for Pediatric Oncology, Utrecht, The Netherlands. [5]Present address: DISCO Pharmaceuticals, Cologne, Germany. [6]These authors contributed equally: Yajie Zhu, Konstantinos Sofiadis. ✉e-mail: argyris.papantonis@med.uni-goettingen.de

## Results

### vU1 snRNA ablation affects hiPSC cell cycle progression

Human vU1 snRNA copies on chr1 (Fig. 1a) are expressed at different levels across cell types and developmental stages. We decided to focus on two such variant genes, vU1.3/.4 (that are too similar in sequence to discriminate) and vU1.8, for three reasons. First, because they are significantly more expressed in embryonic tissues and organs ($P < 0.05$ in Student's $t$-tests), and up to ~20% the levels of canonical U1 snRNA

(Supplementary Fig. 1a). Second, because the expression levels of these vU1s correlate highly with one another in total RNA-seq data from multiple tissues queried, but not with the levels of canonical U1 (Supplementary Fig. 1b), which might hint to non-overlapping functions of the resulting snRNPs. Third, because of their sequence deviation in respect to the canonical U1. vU1.3/.4 carry a C-to-T mutation at the 5' splice site recognition domain, which could allow recognition of AT splice donor sites instead of canonical GT ones

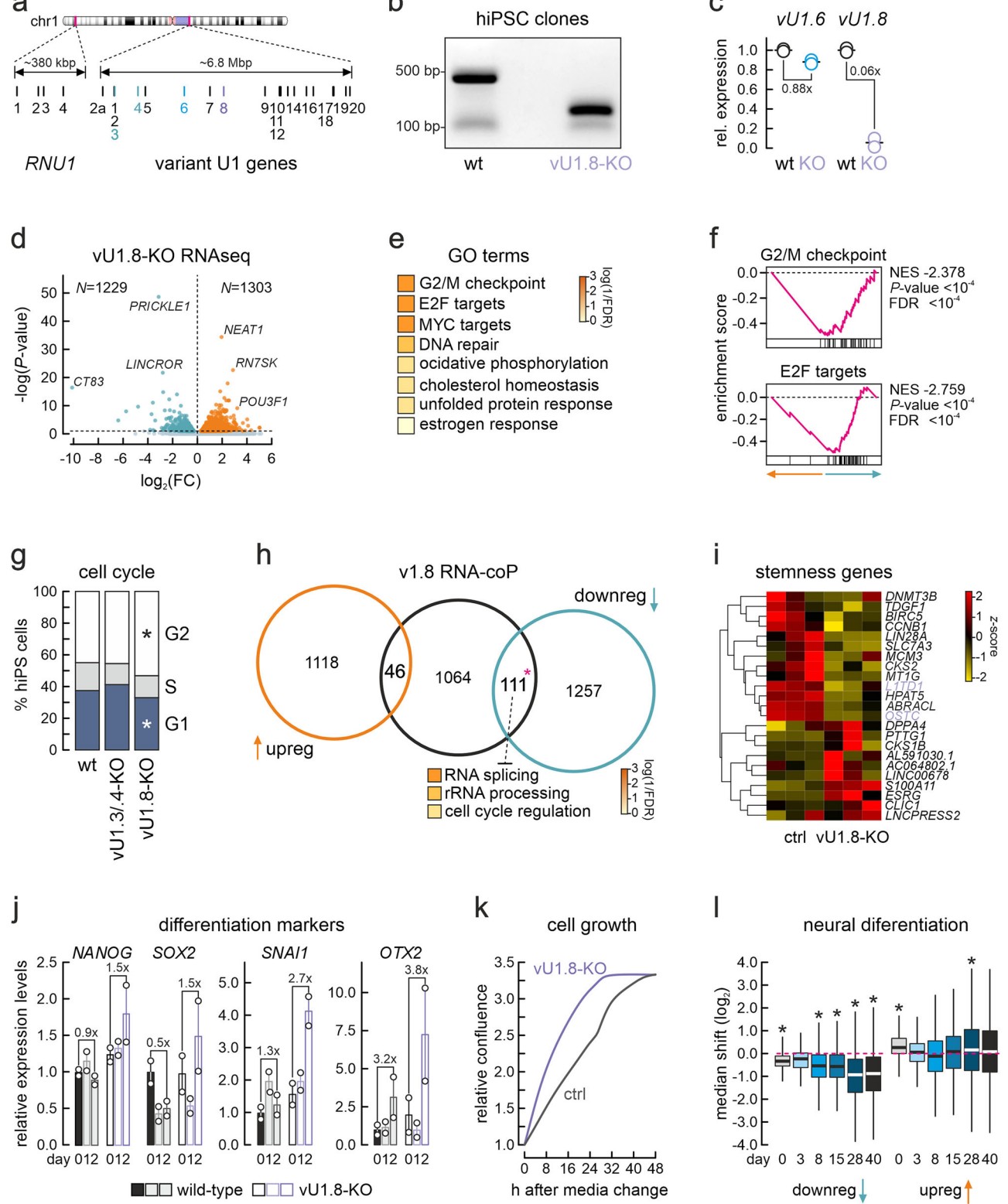

**Fig. 1 | Knocking-out vU1.8 affects gene expression and cell cycle progression in hiPSCs. a** Ideogram of human chr1 (top) with the regions harboring canonical (bottom left) and variant U1 snRNA genes magnified (bottom right), and the vU1.3/.4 (green) and vU1.8 loci (purple) indicated. **b** Representative electrophoretic profiles of PCR products corresponding to the wild-type (wt) and knocked-out *RNU1-8* locus (vU1.8-KO; repeated at least three times in two independent clones). Size marker positions are indicated. **c** RT-qPCR data from two independent replicates showing changes (normalized mean ± SD) in vU1.8 expression in KO relative to wt hiPSCs; vU1.6 levels provide a control. **d** Volcano plot showing differentially up- (orange) and downregulated genes (green) in vU1.8-KO cells, given a $P_{adj}$ cutoff of <0.05. **e** Top gene ontology (GO) terms associated with the downregulated genes from panel (**c**). **f** GSEA enrichment plots for the downregulated genes from panel (**c**). **g** Bar plots showing the percentage of wild-type, vU1.3/.4- or vU1.8-KO hiPSCs per cell cycle phase. *$P < 0.05$, two-tailed Fisher's exact test. **h** Venn diagram showing the overlap of DEGs from panel (**c**) with transcripts co-purifying with vU1.8. *$P < 0.05$, one-tailed hypergeometric test. **i** Heat map showing expression of known stemness-related genes in control (ctrl) and knockout hiPSCs (vU1.8-KO). Genes directly bound by vU1.8 are highlighted (purple). **j** Bar plots showing changes in expression (relative to the mean of day 0 ± SD from two independent replicates) of the indicated genes for control (gray) and vU1.8-KO hiPSCs (purple) at 1 or 2 days of differentiation. **k** Line plots showing changes in relative confluence by control (gray) and vU1.8-KO hiPSCs (purple) in the 48 h after initiating differentiation. **l** Box plots showing changes in expression (log$_2$FC; center line shows the median, boxes indicate the 25$^{th}$ and 75$^{th}$ percentile values, and whiskers extend to 1.5x this interquantile range) for down- (left) and upregulated genes during the differentiation of stem cells into neuronal organoids (using at least three independent replicates per time point; from ref. [29]) relative to their change in vU1.8-KO hiPSCs (using a total of seven independent replicates). *$P < 0.01$, two-tailed Wilcoxon rank-sum test.

(Supplementary Fig. 1c). vU1.8 carries an almost intact 5' splice site recognition domain and a small deletion in its U1-A protein binding site[22]. These vU1 genes also contain multiple other base substitutions, including some in their U1-70K binding domain (Supplementary Fig. 1c) and overall deviate enough so that specific gRNAs could be designed for their genetic ablation (Supplementary Fig. 1d, Supplementary Table 1).

Using CRISPR/Cas9-mediated deletions, we managed to knock out apparently all vU1.8 and nearly all vU1.3/.4 copies in hiPSCs to generate at least two single cell-derived vU1.3/.4-KO and vU1.8-KO clones (Fig. 1b, c, Supplementary Fig. 2a). These KO lines did not exhibit obvious phenotypic differences to wild-type hiPSCs or to one another (Supplementary Fig. 2b). However, each vU1 knockout led to pronounced and non-overlapping effects on hiPSC gene expression as revealed by RNA-seq on at least two independent knockout clones for each genotype. vU1.8-KO produced the strongest effects, with 1303 up- and 1229 down-regulated genes (given a $P_{adj} < 0.05$ cutoff; Fig. 1d). Among the downregulated genes, we found those encoding the key "reprogramming" transcription factors *MYC* and *SOX2*[27], indicative of perturbed stem cell identity. More broadly, gene ontology (GO) and gene set enrichment analyses of downregulated genes revealed a strong association with cell cycle regulation (i.e., "G2/M checkpoint", "E2F targets", "MYC targets"; Fig. 1e, f). Accordingly, cell cycle profiling of PI-stained wild-type and vU1.8-KO hiPSCs revealed accumulation of cells in the G2 phase of the cell cycle at the expense of G1-cells (Fig. 1g, Supplementary Fig. 2c; **gating strategy shown in** Supplementary Fig. 3). This was also corroborated by significantly higher expression of the G2-specific marker CCNB1 in vU1.8-KO compared to wild-type and vU1.3/.4-KO hiPSCs (Supplementary Fig. 2d).

In vU1.3/.4-KO cells, we found an order of magnitude fewer differentially expressed genes (DEGs) with 122 up- and 244 downregulated (given the same $P_{adj}$ cutoff; Supplementary Fig. 2e). GO term analysis showed highest enrichment of upregulated genes for terms like "DNA conformation change" and "RNA metabolism", whereas downregulated ones associated with "response to wounding", "covalent chromatin modification", "chromatin remodeling" or "growth stimulus response" (Supplementary Fig. 2f). Despite the apparent differences in the magnitude and nature of the gene expression changes triggered, these vU1 knockouts converge in that they both deregulate genes involved in nucleosome assembly, RNA metabolism and splicing, as well as in the Notch signaling pathway (e.g. SWI/SNF complex components SMARCA2/4 and ARID1A/B, nuclear speckle components *SRSF2/3/6/7* and *SRRM2*, or Notch pathway components NOTCH1/3; all DEGs listed in Supplementary Data 1) all important for hiPSC maintenance as well as differentiation. Interestingly, 46 genes that are up- and 15 that are downregulated in vU1.8-KO show the converse behavior in vU1.3/.4-KO cells, while only 8 are down in both KOs (e.g., *CCN2*, *FGF2*, *NUS1*), and *HTR7* is the single common upregulated gene.

As is usually the case, knockout lines come with both direct and indirect (compensatory) effects on gene expression. To try and account for these, we cataloged RNAs co-purifying with either the vU1.8 or the vU1.3/.4 snRNAs using the knockout lines as baseline controls, and scrambled oligos in wild-type hiPSCs to control for random noise. Following barcoding and sequencing (see **Methods**), these assays retrieved >1400 RNAs interacting with vU1.3/.4 and >1200 with vU1.8, of which 203 were shared (Supplementary Fig. 2g). Interestingly, RNAs interacting with either vU1 were on average longer than control ones (Supplementary Fig. 2h) and included many mRNAs encoding cell cycle regulators, chromatin assembly factors and remodeling enzymes (e.g. *CCNB1*, *CCNA2*, *SMC3*, *SMARCA2/-C1/-D1, CHAF1A*; see Supplementary Data 2).

We crossed the lists of DEGs from each knockout with the catalogs of co-purified RNAs to find that <11% of vU1.3/.4-KO DEGs qualified as potentially direct targets of these variants (Supplementary Fig. 2i). For vU1.8, this percentage further drops to ~4% for up- and to 8.5% for downregulated genes in vU1.8-KO hiPSCs (Fig. 1h). Nevertheless, the 25 down-regulated vU1.3/.4 putative targets and 111 of vU1.8 are more than expected by chance. In addition, they are enriched for genes associated with "histone modifications", "lipid transport" and "FGF signaling" GO terms in the case of vU1.3/.4 (Supplementary Fig. 2i), and with "RNA splicing", "rRNA processing", and "cell cycle regulation" in the case of vU1.8 (Fig. 1h). Thus, both KOs trigger gene expression changes that, despite a difference in magnitude, suggest a functional role for these vU1s in hiPSCs.

## vU1.8, but not vU1.3 ablation perturbs hiPSC transcriptional homeostasis

The pluripotent nature of hiPSCs allows them to give rise to all cell types in the three major developmental lineages. We therefore asked whether either knockout affects genes linked to stem cell identity maintenance or to the ability of hiPSCs to differentiate. We based this question on our observation of higher vU1 expression in embryonic tissues (Supplementary Fig. 1a), and on data showing vU1 upregulation during cell reprogramming into iPSCs[19].

To this end, we queried the expression of 23 genes considered "stemness" markers for hiPSCs[28]. 13 of these were consistently downregulated in vU1.8-KO cells, while 4 consistently upregulated (Fig. 1i). Of the former 13 genes, *L1TD1* and *OSTC* were also directly bound by vU1.8 (Supplementary Data 2). In vU1.3/.4-KO cells, such changes were only detected for two genes, *DNMT3B* and *SLC7A3*. Next, we tested the response of vU1-KO hiPSCs to undirected differentiation (see **Methods**). We measured gene changes expression at 1 or 2 days after differentiation was induced to find that key stem cell markers like *NANOG* and *SOX2* were induced by ~1.5-fold at day 2 in the absence of vU1.8, along differentiation markers like *OTX2* and *SNAI1* that were also over-induced compared to wild-type hiPSCs (Fig. 1j). Again, no such effects could be observed in vU1.3/.4-KO cells (although baseline expression was slightly higher; Supplementary Fig. 2j). Notably, these expression

changes in vU1.8-KO cells were accompanied by higher growth rates compared to control hiPSCs (Fig. 1k), in line with our RNA-seq and cell cycle data analyses (Fig. 1e–g).

Finally, since the vU1.8-KO led to the deregulation of stemness genes (Fig. 1i) and of TFs involved in neural development (e.g., OTX2; Fig. 1j), we examined how DEGs upon vU1.8-KO compare with the gene expression changes that occur during the generation of neuronal organoids from hiPSCs[29]. Looking at RNA-seq data from 0 to 40 days post-differentiation, we found that genes suppressed in the absence of vU1.8 would normally be downregulated in later neurogenesis stages (days 28–40; Fig. 1l). On the other hand, genes already induced in vU1.8-KO hiPSCs would normally be upregulated within 1 day of differentiation and then again in much later stages (day 28; Fig. 1l). These analyses suggest that the vU1.8-KO drives perturbed hiPSC transcriptional profiles that in part liken those seen later in development.

## vU1 ablation leads to widespread alternative splicing changes in hiPSCs

As U1 snRNAs are essential spliceosome components involved in 5′ splice site recognition, we sought to assess the contribution of vU1.3/.4 and vU1.8 to alternative splicing (AS). AS in stem cells, can lead to inclusion or removal of protein domains encoded by mRNAs, thereby affecting their subcellular localization, coding potential and stability[30]. Thus, we first used IsoformSwitchAnalyzeR[31] to annotate and quantify the most pronounced changes in mRNA isoforms in each vU1-KO line (listed in Supplementary Data 3). vU1.3/.4-KO hiPSCs showed a significant number of isoform changes leading to loss of encoded functional domains ($N = 92$) and to shortening of open reading frames (ORFs, $N = 59$; Fig. 2a, b). On the other hand, vU1.8-KO cells only showed a significant number of changes ($N = 130$) related to loss of ORFs from mRNAs (Fig. 2a, b). For example, ZNF774, a suppressor of Notch signaling[32], and SHC4, a signaling modulator in stem cells[33], both show significant upregulation of non-coding isoforms at the expense of mRNAs with intact ORFs in vU1.3/.4- and vU1.8-KO, respectively (Supplementary Fig. 4a). Still, of the hundreds of genes that undergo detectable isoform changes upon each vU1-KO, only 15 are shared, highlighting their functional divergence (Supplementary Fig. 4b). Of these, PDE4C, SORBS2, KLK8, PCDH11Y and STX1A stand out as they are induced during neuronal differentiation[29].

For a more precise understanding of the AS changes induced after each vU1-KO, we used Whippet[34] to map and quantify AS events from RNA-seq data. We mapped ~6000 and >7500 mRNAs with at least one AS event in our vU1.3/.4- and vU1.8-KO hiPSC lines, respectively. Of these, ~40% involved alternative 3′ end (TE) and 25-30% alternative transcription start (TS) site usage in either vU1 knockout (Fig. 2c). In vU1.8-KO hiPSCs, there were also >20% of mRNAs displaying inclusion/skipping of cassette exons (CE; Fig. 2c). Still, it was not clear how many of these events were a direct consequence of vU1 loss, since each KO also induced or suppressed the expression of various splicing factors (Supplementary Data 1). Therefore, we crossed the full list of AS events (from Supplementary Data 4) with the putative mRNA targets of each vU1 snRNA (from Supplementary Data 2). This narrowed down the list of AS mRNAs to ~700 in vU1.3/.4- and to ~860 in vU1.8-KO hiPSCs, without changing the fact that TS and TE events were again significantly overrepresented in both cases (Fig. 2d). These putative AS mRNA-targets of vU1.8 were shorter and of lower GC content compared to all AS mRNAs, which was not the case for vU1.3/.4 ones (Fig. 2e, f). Moreover, in these presumed vU1.8-targeted mRNAs, reduced 5′ donor site usage was enriched (AD; Fig. 2d), but without noticeable biases in 5′ consensus motifs. Among the 22 AD vU1.8-targets are CPSF7—a key player in pre-mRNA cleavage and polyadenylation[35], ARPP19 and CCDC57—regulators of the G2/M transition[36,37], as well as BAZ2A and SMARCA2—chromatin remodeling complex components linked to stem cell maintenance and proliferation[38–40]. On the other hand, vU1.3/.4 targets showed

enrichment for the exclusion of cassette exons (Fig. 2d), with representative AD/CE events validated via RT-qPCR (Fig. 2g, Supplementary Fig. 4c).

As alternative TE events were the most prevalent in our vU1-KO lines, and the canonical U1 snRNP has been shown to suppress premature 3′-end cleavage and polyadenylation via "telescripting"[41], we analyzed 3′ end AS events further by quantifying 3′ UTR usage with respect to the levels of the last exon in each mRNA[42] (see **Methods**). This showed that the vU1.8-KO led to significantly more 3′ UTR shortening and downregulation compared to both wild-type and vU1.3-KO hiPSCs (Fig. 2h). GO term analysis also showed that genes with shortened and/or less used 3′ UTRs in vU1.8-KO cells were associated to "mitotic cell cycle", "DNA metabolism" and "chromatin organization", as well as to "neural differentiation" (Fig. 2h). One such example is TIMP2, encoding a factor involved in ECM degradation to promote hiPSC renewal[43]. Focusing on mRNAs that co-purified with each vU1 (Supplementary Data 2), the enrichment for shorter and less used 3′ UTRs was corroborated, with 89 mRNAs being regulated in vU1.8-KO hiPSCs (Fig. 2i). A characteristic example is the gene encoding topoisomerase IIα (Fig. 2j) that plays a key role during cell division[44].

Last, we examined the extent to which AS changes were coupled to changes in expression. Intersecting vU1-KO DEGs with genes undergoing AS (Supplementary Data 1, 4) revealed significant overrepresentation only in vU1.8-KO cells and only for CE inclusion and intron retention (Supplementary Fig. 4d). While both increased and decreased intron retention correlated with upregulated gene expression, it was CE inclusion (but not skipping) that predominantly correlated with higher expression levels (Supplementary Fig. 4e). This could be explained by the concept of "exon-mediated activation of transcription"[45], whereby exon inclusion into mRNAs may activate transcription from exon-proximal weak promoters. In fact, when comparing the distribution of CEs related to annotated gene transcription start sites (TSSs), we found that those included more in the transcripts of vU1.8-KO upregulated genes were located significantly closer to TSSs than all other CEs in DEGs (Supplementary Fig. 4f). Similarly, alternative 3′ UTR usage in vU1.8-, but not vU1.3/.4-KO cells, also correlated to differential gene expression (Supplementary Fig. 4g). 134 mRNAs showing increased usage of an alternate 3′ UTRs were significantly upregulated (e.g., TOP2A), while the converse applied to 46 mRNAs with decreased alternate 3′ UTR usage (e.g., CTNNB1 that encodes β-catenin[46]). Notably, the promoters of these 46 mRNAs were enriched for motifs of transcription factors regulating cell cycle progression, like p53, MYC and E2F1 (Supplementary Fig. 4g), in line with the cell cycle effects observed upon vU1.8-KO (Fig. 1f, g).

## Recursive splicing patterns are altered in the absence of vU1 snRNAs

Studies in various organisms, including flies[47,48] and man[49,50], have described an additional splicing layer whereby longer introns may be removed in a stepwise manner rather than via a single splicing reaction. This was accordingly named "recursive splicing" (RS) and involves the splicing of 5′ donor sites into cryptic sites located deep inside of introns (reviewed in[51,52]). In flies, this occurs at YAG|GU "zero-length" exons[47]. In man, however, we documented intronic sites producing RS intermediates that did not always contain a conventional GU donor sequence. In fact, the resulting RS donor was not a GN dinucleotide in >55% of cases[49]. Given that vU1 snRNAs have 5′ ss domains that deviate from the canonical one, we wondered whether they might also be involved in the recognition of specific RS site subsets in hiPSCs.

To address this question, we first mapped hybrid reads representing RS events (i.e., 5′ ss donors at the end of exons spliced into YAG|NN intronic sequences[49]) across a collection of ENCODE total RNA-seq data[53]. This analysis showed that RS is more prevalent in cell types of embryonic origin (Fig. 3a), matching the expression profiles

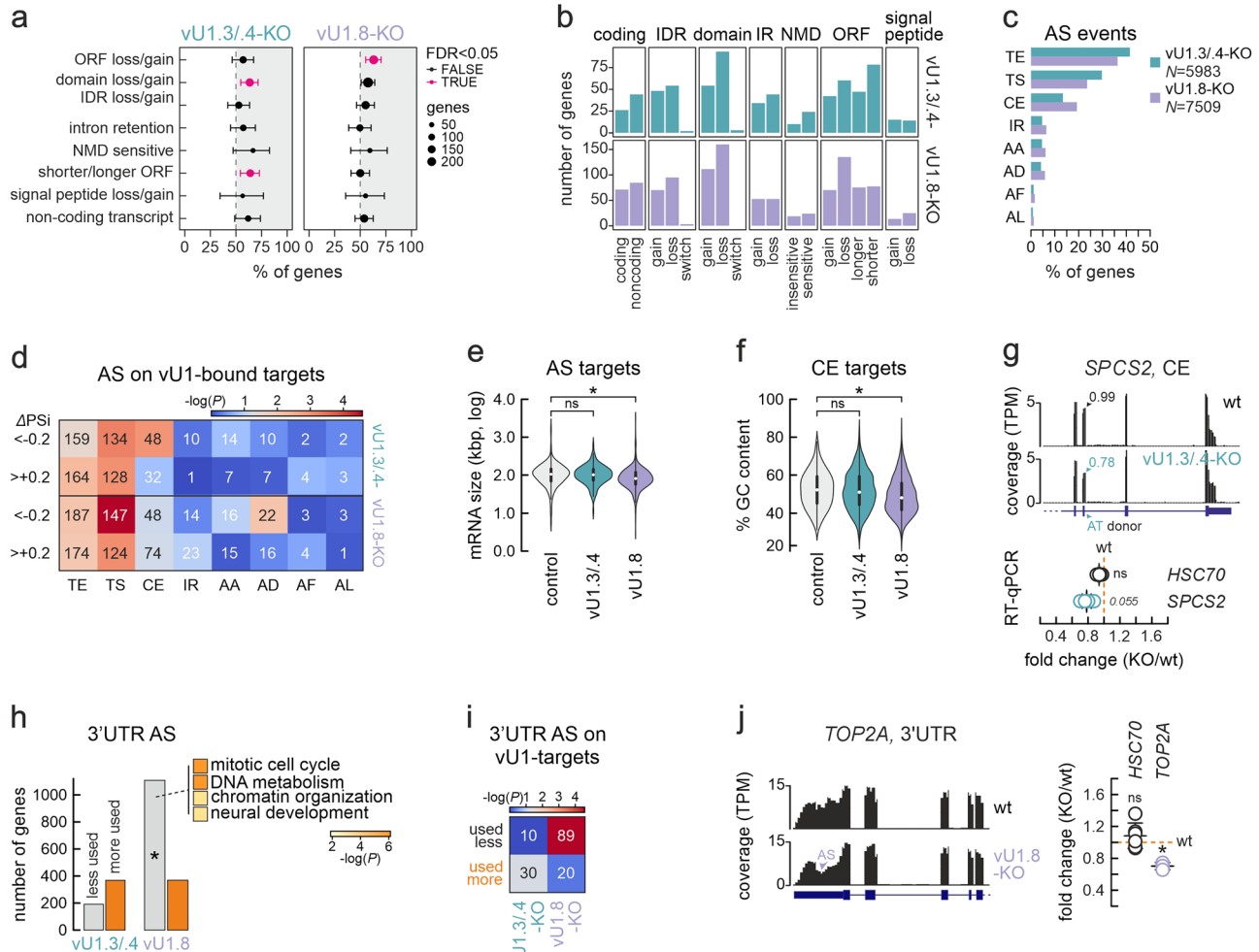

**Fig. 2 | Effects of vU1-KOs on hiPSC alternative splicing patterns. a** Plot showing the percent of alternatively spliced genes (mean ±95% confidence intervals) that show each of the listed mRNA isoform changes with respect to all gene-changing isoforms between wild-type and vU1-KO hiPSCs. Changes that occur significantly more often (FDR < 0.05; using at least three independent replicates per genotype) are indicated (magenta). **b** Bar plots showing the number of genes that show any of the indicated isoform changes in vU1.3/.4- (top) or vU1.8-KO hiPSCs (bottom). **c** Bar plots showing the percentage of genes exhibiting at least one of the indicated AS changes in vU1.3/.4- or vU1.8-KO hiPSCs. The total number of AS genes (*N*) is indicated. TE, alternative transcription end site; TS, alternative transcription start site; CE, alternative core exon usage; IR, intron retention; AA, alternative acceptor site; AD, alternative donor site; AF, alternative first exon; AL, alternative last exon. **d** Heat map showing the number of events per AS type for vU1-associated transcripts in vU1.3/.4- (top) or vU1.8-KO hiPSCs (bottom). The color code reflects statistical overrepresentation (-logP) in each case. **e** Violin plots (center shows the median, rectangles indicate the 25th and 75th percentile values, and the plots extend to 1.5x this interquantile range) showing the length distribution of alternatively spliced vU1.3/.4- (green) and vU1.8-bound mRNAs (purple) compared to all AS

transcripts (gray) using at least three independent replicates per genotype. *P < 0.01, two-tailed Wilcoxon-Mann-Whitney test. **f** As in panel (**e**), but **f** or the mean GC content of AS cassette exons in these mRNAs using at least three independent replicates per genotype. *P < 0.01, two-tailed Wilcoxon-Mann-Whitney test. **g** Top: Genome browser view of RNA-seq coverage along the *SPCS2* gene from wild-type and vU1.3/.4-KO hiPSCs. The percent usage of the indicated AT donor site is shown. Bottom: RT-qPCR validation (mean ± SD from at least three independent replicates per condition) of this AS event in VU1.3/.4-KO hiPSCs (green) relative to wild-type levels (orange dotted line); splicing of *HSC70* exons provides a control. *P < 0.01, unpaired two-tailed Student's *t*-test. **h** Bar plots showing the number of genes with **h** reduced (gray) or increased usage of alternative 3'UTRs (orange) in vU1.3/.4- (left) and vU1.8-KO hiPSCs (right). *P < 0.01, two-tailed Fisher's exact test. The four most enriched GO terms for the mRNAs showing reduced 3'UTR usage are shown. **i** As in panel (**d**), but for alternative 3'UTR usage of vU1-bound targets in each KO line. **j** As in panel (**g**), but for alternative 3'UTR usage in the *TOP2A* gene (mean ± SD from at least three independent replicates per genotype). *P < 0.01, unpaired two-tailed Student's *t*-test.

of vU1s (Supplementary Fig. 1a). However, as many of these RS events might lead to non-productive splicing (as discussed in[52]), we sought to obtain more confirmatory data. First, we selected two exemplary RS intermediates not involving a canonical GT in their RS donor site, and used CRISPR/Cas9 to specifically delete either RS site in HEK293 cells and assess their contribution to mRNA production from their host gene. Deletion of the RS site in the first intron of *UBE2E2* led to >50% reduction in mRNA levels, whereas that in the first intron of *UQCC1* had no discernible effect (Supplementary Fig. 5a). This is in line with our previous testing of consecutive RS sites within the first long intron of *SAMD4A* in human endothelial cells[49]. Second, we

looked into RS attributes across cell types. Collectively, there appears to be a bias for RS occurring in genes encoding DNA- /RNA-binding proteins, enzyme modulators or transcription factors (Fig. 3b). RS occurrence was not exclusive to long introns, but their numbers did scale with the median length of the introns in which they occur (Fig. 3c). We also stratified RSSs into conserved (Phast-Cons score >0.45 across vertebrates) and non-conserved ones (score <0.45; Fig. 3d), but found no bias in sequence composition between them (Fig. 3e) showing that RSSs with canonical donors are not more conserved and, thus, cannot be presumed as functionally more relevant.

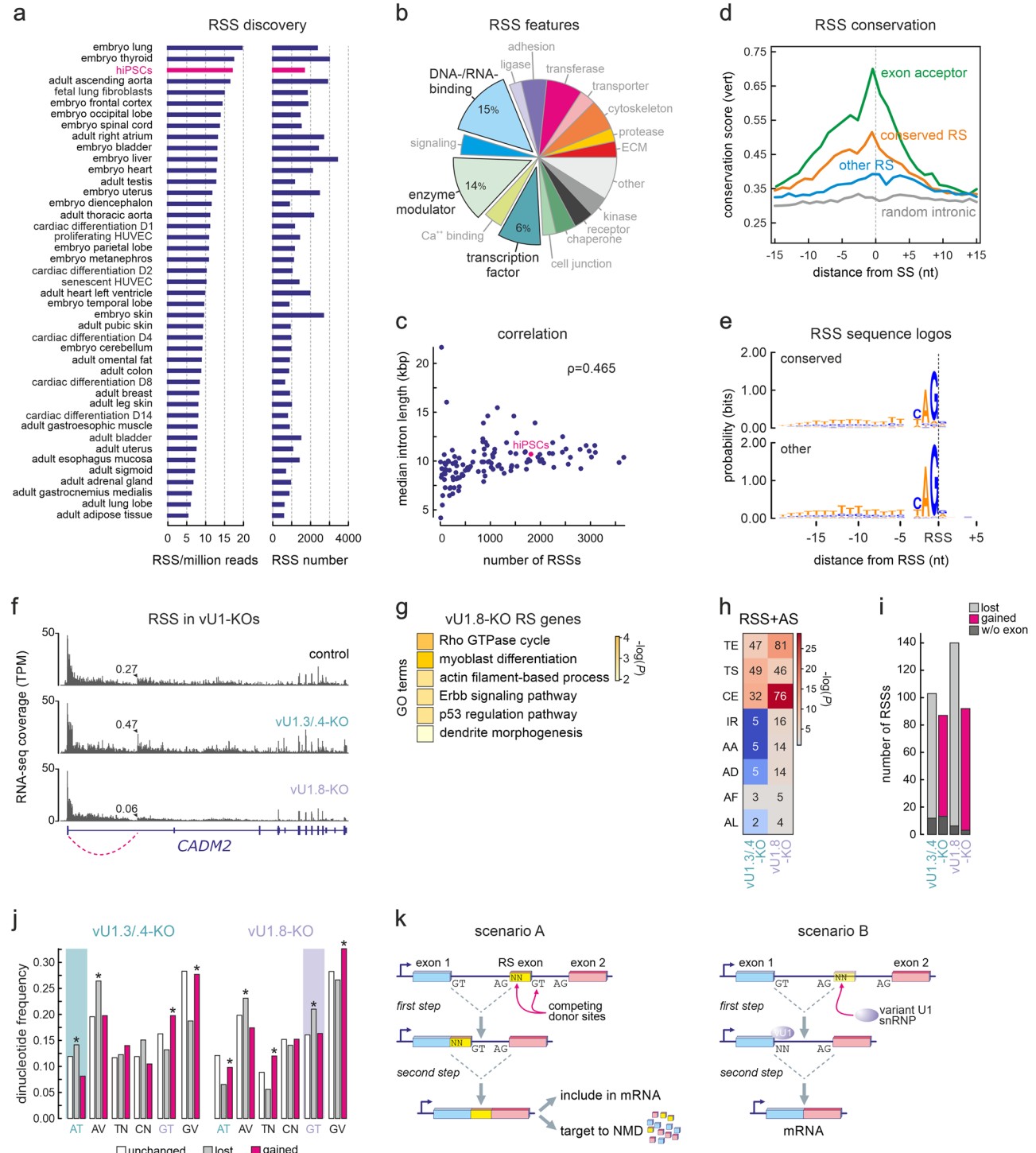

**Fig. 3 | vU1 contribution to recursive splicing in human cells. a** Bar plot showing the number of recursive splicing sites (RSSs) per million RNA-seq reads (left) and their total numbers for each tissue or cell type. **b** Pi chart showing the percent of genes carrying RSSs from panel a that fall into different biological function GO categories. The top three categories are highlighted. **c** Plot showing the correlation between total RSS numbers and median intron length for each cell type from panel (**a**). The Pearson's correlation coefficient (ρ) is indicated. **d** Line plot showing the conservation score (from all vertebrates in the UCSC browser) in the 30 nt around canonical 3' acceptor sites (green), RSSs with high ( > 0.45; orange) or low conservation ( < 0.45; blue), and randomly selected sites from the same introns (gray). **e** Logos showing sequence probability at each position around highly ( > 0.45) and lowly conserved RSSs ( < 0.45) from the data in panel a. **f** Genome browser view of RNA-seq coverage from the *CADM2* gene showing usage of an exemplary RSS

(arrowhead) in wild-type, vU1.3/.4-, and vU1.8-KO hiPSCs. **g** Top GO terms associated with recursively spliced genes in vU1.8-KO hiPSCs. **h** Heat map showing the number of RSSs linked to AS events in vU1.3/.4- (left) or vU1.8-KO hiPSCs (right). The color code reflects statistical overrepresentation (-log*P*) in each set. **i** Bar plots showing the number of RSSs lost (gray) or gained (magenta) in vU1.3/.4- (left) and vU1.8-KO (right) compared to wild-type hiPSCs. The number of RSSs not associated with an immediate downstream exon in each case is indicated (dark gray). **j** As in panel (**h**), but for specific RS dinucleotides (N: A, T, C, or G; V: A, C, or G). The RS donor dinucleotides matching vU1.3/.4 (left) and vU1.8 5'ss sequences (right) are highlighted. *\*P* < 0.05, two-tailed Fischer's exact test. **k** Two scenarios for how RS-exon inclusion in mRNA via competition between RS and canonical donor sites may lead to the production of translatable or NMD-prone transcripts (scenario A) or for the inclusion/exclusion of cryptic exons (scenario B).

Third, we hypothesized that if RS sites are indeed functionally relevant, they may contain single-nucleotide polymorphisms (SNPs) linked to diseases or traits in a cell type-specific manner. Based on the fact that 70-90% of SNPs associated with common complex diseases and traits occur in non-coding parts of chromosomes, including introns, we previously developed GARLIC, a software that etiologically associating putative disease-causative genetic variants with regulatory sequences of interest[54]. GARLIC is linked to a curated SNP database drawing from genome-wide association studies, and we used it to assess statistical enrichment of genetic variants in the 25 nt around RS sites (i.e., 5 nt down- and 20 nt upstream of each RSS to include associated T-tracts). Initially, we used a catalog of RSSs from a time course of hiPSC differentiation into cardiomyocytes[55] along which we saw a gradual decrease of RS events (Supplementary Fig. 5b). However, SNPs associated with cardiomyopathy and echocardiogram traits were specifically enriched in cardiomyocyte RS sites (not in earlier progenitors; Supplementary Fig. 5b). Thus, we expanded this testing to the whole GARLIC database using RS catalogs from all available ENCODE cell types. This revealed various clusters of relevant cell types and traits, e.g., RS sites from neural cells being enriched for SNPs associated with neurodegeneration, depression, migraine, psychosis and bipolar disorders (Supplementary Fig. 5c). These associations provide further support to the notion that RS is likely consequential for gene regulation.

Given the pervasive presence of non-GT donors at RS sites and the fact that hiPSCs display high RS occurrence, we asked whether vU1 snRNAs can be implicated in RS implementation (Fig. 3a). To increase stringency here, we applied additional filtering criteria for RS events (adapted from ref. 56; see **Methods**) and quantified RS usage by normalizing junction reads containing an RS site to all junction reads that include its upstream exon. This way, we can statistically determine the over- or underrepresentation of a given RS event in vU1-KO cells compared to control hiPSCs (see example in Fig. 3f). In total, we identified ~200 and >230 differentially used RS sites in vU1.3/.4- and vU1.8-KO cells, respectively. Notably, genes with differentially used RSSs in vU1.8-KO cells were associated with GO terms like "Rho GTPase cycle", "p53 regulation" and "Erbb signaling" relevant to cell cycle progression and survival (Fig. 3g).

Using these RSS catalogs, we first asked whether differentially used RSSs correlated with alternative splicing. Crossing AS events with RS genes in each KO line showed that vU1.3/.4 loss correlated equally with alternative TS and TE, as well as cassette exon usage, whereas vU1.8 loss produced a similar profile but with a stronger enrichment for CE usage (Fig. 3h). In fact, when it comes to the location of these alternatively included/skipped CEs, we found that the differential use of an RS site predominantly affected exons downstream of that site, implicating recursive splicing in mRNA composition outcomes (Supplementary Fig. 5d).

Looking more into these differentially gained or lost RS sites in our hiPSC lines, we saw that consistently more are lost than gained upon vU1-KO (Fig. 3i). However, we wanted to know whether hiPSC RS sites were "zero-length" exons (as in Drosophila[48]) or involved an RS-exon (as proposed for human neural cells[50]). RS-exons are thought to provide "exon definition" to RS sites, but are not included in mature transcripts[50]. In our KO lines, the vast majority of RS sites (89% or more) are followed by an RS-exon (Fig. 3i). However, a non-negligible fraction of these RS-exons display signal enrichment in hiPSC RNA-seq data, meaning that they likely represent cryptic exons, the usage of which is under vU1 control (Supplementary Fig. 5e). Finally, we asked to what extent the loss of a vU1 would lead to selective loss or gain of RS usage based on the RS-donor motif that the variant snRNAs might recognize. For vU1.3/.4-KO cells, we recorded a significant decrease in the usage of RS sites with AN donors with concomitant increase in GN usage (Fig. 3j). Conversely, vU1.8-KO cells displayed a significant drop in the usage of GT RS-donors with an increase in GV and AT ones (V = A/

C/G; Fig. 3f, j). These effects agree with the theoretical ability of vU1.3/.4 to recognize AT donors, which is not the case for vU1.8 (Supplementary Fig. 1c). Notably, we saw no bias in the differential loss or gain of conserved (PhastCons score > 0.45) over non-conserved RS sites (score < 0.45) in either KO, while only the introns where RS events were gained upon vU1.8 were significantly different in size to control ones (Supplementary Fig. 5f). Together, our analyses suggest that vU1s can affect RS patterns genome-wide in specific subsets of introns.

## Discussion

Even though variant U1 snRNAs are most expressed in pluripotent cells, their expression quickly declining upon differentiation[18], their functional roles in gene expression control remain understudied. This is due to the predominance of canonical U1 snRNA expression (vU1.3/.4 and vU1.8 express at <1/4 the levels of U1 in the cell types we analyzed) and the sequence redundancy among the many vU1 gene copies. We therefore selected vU1 genes, *RNVU1-3/-4* and *-8*, that could be specifically deleted from chr1 of hiPSCs in order to study the consequences of their loss on stem cell maintenance and gene expression.

Using these hiPSC knockout lines, we found that the ablation of either vU1 snRNA leads to profound gene expression changes, albeit to different extents (vU1.8-KO produces an order of magnitude more DEGs than vU1.3/.4-KO). However, most of these gene expression effects appear to be indirect, as they only concern a subset of vU1 target RNAs (<13% and <3% for vU1.8- and vU1.3/.4-KO targets, respectively). At the same time though, and as might be expected of snRNAs that can assemble into apparently functional snRNPs[24], both knockouts trigger strong changes in hiPSC splicing patterns, with hundreds of their targets exhibiting alternative usage of transcription start/end sites and cassette exons. This finding corrects previous postulations (based on mini-gene reporters) of limited vU1 contribution to splicing[57]. We also asked what the implications of these changes to hiPSC homeostasis are. These seem to primarily manifest in vU1.8- rather than vU1.3/.4-KO cells, likely reflecting their difference in effect magnitude. Despite the fact that no apparent morphological differences were observed, the loss of vU1.8 resulted in cell cycle changes, with KO-hiPSCs exhibiting a shorter G1 phase and faster growth. This may be in part explained by the *MYC* transcript constituting a direct vU1.8 target that is both downregulated ($\log_2 FC = -0.7$) and alternatively spliced at its TSS in knockout cells. In the absence of vU1.8, key transcription factor genes like *SNAI1* and *OTX2* also exhibited aberrant regulation upon non-directed differentiation, in agreement with the broad and early changes in the expression of many genes known to be differentially regulated during a neuronal differentiation time course, including those in the critical Wnt and Notch pathways. Interestingly, there exist reports linking the deletion of the chromosome region (1q12-21) that harbors most vU1 snRNA genes to severe pediatric neurological dysfunctions[58], while motor neuron cultures derived from SMA patients sustained abnormally high vU1 expression[19]. Both these examples further link vU1 regulation to proper neurodevelopmental progression.

Finally, we entertained the hypothesis that vU1 snRNAs, due to their potential for pairing to variable 5′ splice site dinucleotides, might also be involved in the regulation of recursive splicing[52]. Here, we recorded a bias in the usage of AT and GT 5′ ss RS-donors between vU1.3/.4- and vU1.8-KO cells, despite the fact that high redundancy in RS site recognition and usage is to be expected[49]. Interestingly, the absence of either vU1 led to changes in RS patterns that could be linked to the alternative splicing of the corresponding transcripts. This might be, in turn, connected to the fact that the majority of RS sites are followed by small cryptic RS-exons. These RS-exons are likely competing for inclusion into mRNA with their flanking exons, but many include sequences that target them for degradation (as has been proposed for neuronal cells[50]). Our data suggest that these RS sites can signal the skipping of such RS-exons in the presence of specific vU1s,

providing another layer of RNA quality control (see model in Fig. 3k). Still, a non-negligible number of RS events likely contributes to the stepwise removal of introns and, thus, to the production of mature transcripts. This, alongside the various traits of RS-containing transcripts (e.g., the fact that they often overlap cell type-specific disease-associated SNPs), supports a functional role for this regulatory layer of splicing in human biology and disease.

Taken together, this work assigns vU1 snRNAs with a far more decisive role in human stem cell function and maintenance than previously appreciated. Nevertheless, this now begs a number of new questions. For example, long-read sequencing can be used to investigate the coincidence or not of many of these vU1-dependent alternative splicing effects on single transcripts. Similarly, despite having cataloged vU1.3/.4- and vU1.8-target transcripts, we still lack high-resolution information on their exact positions of interaction on RNA. Last, more functional testing of the contribution of RS in mRNA production is needed, as is a broader screen of the precise contribution of other vU1s to recursive splicing.

## Methods
### Cell culture
Human induced pluripotent stem cells (hiPSCs; GM24581*B from Coriell) were derived from the preprogramming of human GM02036 fibroblasts by overexpressing the four Yamanaka factors (i.e., hOCT3/4, hSOX2, hKLF4 and hL-MYC) via episomal vectors and validated for genomic integrity and their pluripotent character. hiPSCs were grown in StemFlex media and dissociated using Accutase (Sigma-Aldrich) at 37 °C for 10 min when confluent. For non-directed differentiation, hiPSCs were seeded on Matrigel-coated plates in StemFlex Medium with 2 µM Thiazovivin at a density of $4.7 \times 10^4$ cells/cm² for confluence analysis or $2.1 \times 10^4$ cells/cm² for RNA extraction. The following day, the medium was replaced by DMEM (Gibco) supplemented with 10% FBS (Capricorn Scientific) and 2 mM Glutamine (Thermo Fisher) and cultured for 1 to 3 days, with daily medium changes. For confluence analysis, plates were imaged every 30 min for 48 h following differentiation induction in an IncuCyte platform (Sartorius), and confluence was quantified using the Incucyte AI Confluence Analysis™ workflow.

### CRISPR/Cas9 gene knockout
For generating vU1-knockout iPSCs, WT cells were genome-edited using ribonucleoprotein-based CRISPR/Cas9 with crRNA/tracrRNA and Hifi SpCas9, targeting upstream and downstream of each locus via tailored gRNAs (listed in Supplementary Table 1). 300 pmol of Alt-R CRISPR-Cas9 crRNA and 300 pmol of Alt-R CRISPR-Cas9 tracrRNA were pre-assembled with 122 pmol Alt-R Hifi SpCas9 Nuclease 3NLS (all IDT DNA Technologies) to form ribonucleoprotein complexes, and were nucleofected in to $2 \times 10^6$ early-passage ($p < 20$) iPSCs using the 4D Amaxa Nucleofector system (Lonza; program CA-137) and the P3 Primary Cell 4D-Nucleofector X Kit (Lonza) according to the manufacturer's instructions. Following nucleofection, iPSCs were replated into a Matrigel-coated (growth factor-reduced, BD Biosciences) 6-well plate containing StemFlex medium supplemented with 2 µM thiazovivin (Merck) and 100 U/ml penicillin and 100 µg/ml streptomycin (Thermo Fisher). After 3 days, transfected iPSCs were singularized using the CellenOne dispenser (Cellenion/Scienion) in StemFlex medium into Matrigel-coated 96-well plates. Successful genome editing was identified by Sanger sequencing, and several homozygous knockout iPSC lines were established. Generated iPSC lines were maintained on Matrigel-coated plates, passaged every 4–6 days using Versene solution (Thermo Fisher) and cultured in StemMACS iPS-Brew XF medium (Miltenyi) supplemented with 2 µM thiazovivin on the first day after passaging with daily medium changes. All PCR primers used for validating the CRISPR/Cas9 deletions are listed in Supplementary Table 2.

### Reverse-transcription quantitative PCR analysis
Total RNA from control and differentiated hiPSCs was treated with TURBO DNase (Invitrogen) to remove genomic DNA contaminants, and reverse-transcribed using the SuperScript™ IV Reverse Transcriptase (Invitrogen) protocol and random hexamers in the presence of RiboLock RNase Inhibitor (Thermo) in each reaction. The resulting cDNA was used in real-time quantitative PCR (qPCR) reactions using the qPCRBIO SyGreen® Mix Separate-ROX system (PCR Biosystems) as per the manufacturer's instructions in 384-well PCR plates and a final reaction volume of 10 µL. A qTower 384 G (Jena Biosciences) PCR machine was used with the following program: 95 °C for 1 min, then 40 cycles of 95 °C for 15 s and 60 °C for 30 s, followed by melting curve analysis with incremental temperature increase from 60 °C to 100 °C. The expression levels of amplified genes were normalized to the mean expression of GAPDH (for changes in mRNA levels) or of YWHAZ (for AS changes). All primers used are listed in Supplementary Table 3.

### RNA sequencing and analysis
Total cell RNA was TRIzol-extracted from control and vU1-KO hiPSCs as per manufacturer's instructions (Invitrogen), reverse-transcribed using random primers and ribodepleted before being sequenced on a Novaseq6000 platform (Illumina). Paired-end reads from each replicate were mapped to the reference human genome (build hg38) using STAR v.2.7.3a with default settings[59]. Mapped reads were quantified using htseq-count2 v.0.12.4[60] and normalized via the RUVs function of the RUVseq package[61]. For differential expression analysis, vU1-KO and control were compared using DESeq2 v.1.34.0 and the default Wald test[62]. Genes with an adjusted $P$-value $< 0.05$ and fold change ($\log_2$) $> |0.6|$ were considered significantly differentially expressed and are listed in Supplementary Data 1. Functional enrichment analysis of gene sets was performed using Metascape[63] or Gene Set Enrichment Analysis (GSEA; https://www.gsea-msigdb.org/).

### Splicing analyses
Individual RNA splicing events were detected in RNA-seq data, quantified, and compared between control and vU1-KO cells using Whippet[34] with a probability cutoff of >0.9 and a |Δpsi| of >0.2. All AS events detected are listed in Supplementary Data 4. Changes to mRNA isoforms were annotated and quantified via IsoformSwitchAnalyzeR with default parameters[31]. The genes undergoing isoform switching in each vU1-KO are listed in listed in Supplementary Data 3. Recursive splicing sites (RSSs) were first mapped using a custom in-house script to identify hybrid RNA-seq connecting the 3' end of a given exon with downstream sequences in the following intron[49]. Next, RSSs were further filtered and quantified by considering three necessary attributes: identification of an RS motif with a PhastCons score of >0.5, junction identification supported by >5 nt, and identification of a "sawtooth" RNA-seq coverage pattern with >two-fold signal difference between the upstream and downstream regions of the RS site coupled to a permutation test ($P < 0.01$) to evaluate the consistency of signal density difference before and after the candidate RS site[56], while all YAG|NN sequences though (instead of the AG|GT ones considered by the original algorithm) and filtering out any RS events that were also detected in hiPSC poly(A)-enriched RNA-seq data. The number of reads supporting each RS event that fulfills all aforementioned criteria was normalized to all reads connecting the exons flanking the intron containing the RSS. A Student's $t$-test with a $P$-value cutoff of <0.05 was used when comparing conditions. 3' UTR alternative usage was quantified via RNASeq3USP (https://github.com/christear/RNASeq3USP) and the same statistical comparison as above.

### vU1 RNA co-purification and analysis
For the co-precipitation of RNA interacting with vU1 sRNAs, we used 25 pmoles of a biotinylated 2'-O-Methyl RNA antisense oligonucleotide targeting vU1.3 or vU1.8 and ~2 µg of iPSC total nuclear RNA. In brief,

hiPSCs grown in 10-cm plates were crosslinked with psoralen as follows: AMT was first diluted in water at 1 mg/ml and then an equal volume of 2x PBS was added to produce a final concentration of 0.5 mg/ml, which was kept chilled on ice and in the dark at all times. Cells were washed once in 20 ml of ice-cold 1x PBS, and then gently scraped into a 10-ml tube, centrifuged at 420 x g for 5 min at 4 °C, and collected as a pellet. Following isolation of nuclei on 'sucrose cushion', hiPSC nuclei were resuspended in 4 ml of ice-cold AMT solution (or in ice-cold PBS alone that serves as the non-crosslinked control) and incubated on ice for 15 min, before being transferred to pre-chilled 10-cm cell culture dishes. The dishes, kept on ice, were placed under a 35 nm UV bulb in a Stratalinker 2400 (Stratagene) ~ 3–4 cm from the light source for a total of 7 min at maximum power (while mixing gently every 2 min). Irradiated nuclei are then transferred to chilled tubes and spinned at 330 x g for 4 min, before pellets are treated with TRIzol (Invitrogen) to isolate total nuclear RNA and resuspended in 50 µl of water. The RNA yield is quantified (e.g., on a NanoDrop device) and for each 8 µg purified, a 50-µl reaction with 2.5 units of TURBO DNase (Invitrogen) is set up to degrade any genomic DNA contaminants and incubated at 37 °C for 20 min. RNA in each reaction is re-purified using the miRNeasy kit (Qiagen) and eluted in 30 µl of water. Finally, 2.5 pmol of biotinylated 2'O-Methyl vU1-antisense oligonucleotide is added to 2 µg of nuclear RNA (for reference, for U1 snRNAs 20 pmoles are used) after denaturation of the oligonucleotide at 85 °C for 3 min and transferring to ice. The mixture is supplemented with 300 µl of pre-warmed hybridization buffer with LiCl, transferred to 37 °C and incubated under shaking at 1200 rpm for 2 h. The cross-linked vU1:RNA complexes are pulled-down using 20 µl of Streptavidin C1 magnetic beads per reaction, which have first been spun at 12,000 x g at 4 °C to pellet debris. The beads and supernatants are incubated under shaking at 1200 rpm, and then separated using a magnetic tube rack. The beads are last washed 3x in 250 µl of Low Stringency Wash Buffer at 37 °C for 3 min/wash, and 3x in Low Stringency Wash Buffer the same way, before direct immersion in TRIzol to purify the co-precipitated RNAs. To control for unspecific RNAs that may be pulled-down by our targeting oligos, we also performed IPs using the vU1-KO cells matching each oligo; these yielded essentially no RNA, and no cDNA libraries could be made. To control for general background noise in our IPs, non-targeting 'scrambled' oligos were used on wild-type hiPSCs, which could yield cDNA libraries.

vU1 pulldown targets were cataloged using RNA-seq. Due to low RNA yields, the eluates from three independent co-precipitations were pulled together and reversed transcribed using the SMART-Seq Total RNA Pico Input with UMIs kit (ZapR Mammalian; TAKARA Bio, 634354). Resulting libraries retain strand information and incorporate unique molecular identifiers (UMIs) during the reverse-transcription step to correct for PCR biases and assist transcript quantification. Following sequencing on a NovaSeq6000 platform (Illumina), vU1 pulldown targets were quantified with NOISeq[64] under default settings, and those that passed the 0.05 statistical significance cutoff are listed in Supplementary Data 2.

**Immunofluorescence**

$5 \times 10^4$ iPSCs were seeded onto Matrigel-coated coverslips in 24-well plates containing STEM Flex + STEMFLEX Supplement media. 36 h post-seeding, the media was removed, and cells were washed once in 1x PBS (Sigma, D8537), before being fixed with 4% PFA in 125 mM HEPES for 10 min at RT. After removing the fixative, cells were washed 2x in PBS and permeabilized via 3x washes (5 min each) in PBS supplemented with 0.3% w/v Triton X-100 (Sigma-Aldrich, T8787-100ML). Next, cells were incubated for 30 min at RT in a blocking buffer consisting of PBS-T plus 3% w/v BSA (Roth, #T844.2). Finally, the blocking buffer was removed, and cells were incubated with a mouse monoclonal antibody against CCNB1 (Cell Signaling Technology, 4135; dil. 1:400) in PBS-T plus BSA at 4 °C overnight. Next day, cells were washed

with 3x PBS-T for 10 min each and incubated in PBS-T plus BSA with anti-Cy3 goat anti-mouse IgG (Abcam, ab97035; dil. 1:500) for 1 h at RT in a dark chamber. After washing again 3x in PBS-T for 10 min, coverslips were mounted onto slides using ProLong™ Gold Antifade with DAPI (Invitrogen, P36931) before images were acquired using a wide-field fluorescence microscope (Zeiss AxioImager M1).

**Statistical analyses**

All statistical tests were performed in R, except for Student's t-tests and Fisher's exact tests, which were performed via the online version of GraphPad (https://www.graphpad.com/quickcalcs/). Unless stated otherwise, results were deemed statistically significant once a P-value cutoff of <0.01 was met.

**Reporting summary**

Further information on research design is available in the Nature Portfolio Reporting Summary linked to this article.

## Data availability

The RNA-seq data generated in this study have been deposited in the NCBI Gene Expression Omnibus repository under the accession number GSE305587. Source data are also provided with this paper. Source data are provided with this paper.

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

## Acknowledgments

We thank Niels Gehring, Volker Boehm, Maria-Patapia Zafeiriou and Leo Kurian for discussions, and Dawn O'Reilly for the U1-/vU1-targeting oligos and her feedback in the earlier stages of this project. This study was supported by the German Research Foundation (DFG) via the Priority Program 1935 (Project No. 313408820 to A.P.), by the Lower Saxony Ministry for Research and Culture (MWK) via the SPRUNG program (Project No. 76211-1267/2023 to A.P.) and by the Scientific Core Facility for Cell Sorting of the UMG (DFG Large Equipment Project No. 442249343). Y.Z. was supported by the IMPRS Molecular Biology program, and M.A. by a scholarship of the Göttingen Promotionskolleg for Medical students funded by the Jacob-Henle-Program and the Else Kröner Fresenius Stiftung ('Promotionskolleg für Epigenomik und Genomdynamik'; Project No. 2021_EKPK) as well as by a Friedrich-Naumann-Stiftung scholarship from the Federal Ministry for Research, Technology and Space. We acknowledge support by the Open Access Publication Funds of the University of Göttingen.

## Author contributions

Y.Z. performed all computational analyses; K.S., A.M., and M.A. performed experiments; L.C. generated the vU1.8-KO hiPSC lines; M.N. performed GARLIC analyses; V.K. and C.F. performed differentiations; A.P. conceived the project; Y.Z. and A.P. wrote the manuscript with input from all authors.

## Funding

## Competing interests

The authors declare no competing interests.
