## [Transparent Peer Review file · Nature Communications]

Variant U1 snRNAs contribute to cell cycle and differentiation control of human iPS cells

Corresponding Author: Professor Argyris Papatonis

Version 0:

Reviewer comments:

Reviewer #1

(Remarks to the Author)

The manuscript by Zhu and Sofiadis et al. describe analysis of the impact of stem cell knockout of two variants of the U1 snRNA (U1-3 and U1-8). Using cell biology and genomics, they reveal interesting specific effects of the individual knockout of these two U1 variants that have significant expression in stem cells, and show data suggesting these variants play particularly interesting roles in stem cell biology and potentially control unique recursive splicing sites.

To me, the RNA IP experiment is confusingly described – it's described as 'using the knockout cells as baseline controls', but I'm unclear what that means. Fig. 1F has the control as a 3rd experiment, but then I'm not clear what cells are used for that control – is it a vU1.3/vU1.8 double KO? Or a mixture of the two single-knockouts? I also don't think the analysis here is well-described – the methods simply states "vU1 pulldown targets were quantified with NOISeq", but I don't see any sort of mention of statistical tests or even what the criteria is for defining an RNA as 'interacting' (the results just says '>1400 RNAs interacting with...', with the genes simply listed in Table S3, so unless I'm missing something I can't find any discussion of what cutoffs or statistics were used to identify those 1400).

I would suggest that the next paragraph ("We crossed the lists of differentially regulated gene...") be toned down – to me it's unclear whether the data as described can be used to make statements like 'only 10% of DEGs were direct targets'. Trivially, every intron-containing gene will at some level be bound by U1 in the cell (and likely most others, due to U1's telescripting activity); but U1 isn't profiled here to provide a baseline of how easy it is to detect these interactions, and with only 1 replicate each of vU1.3 and vU1.8, I'm not sure it's possible to confidently determine to what degree this pulldown has saturated discovery of vU1.3 and vU1.8-interacting RNAs, or whether the overlaps here reflect just the technical difficulty in pulling down and sequencing U1-interacting RNAs.

I think the RNA IP -> proteomics experiment described in Fig. S2H-I likely needs some degree of replication or validation before publication. The straightforward implication of S2I would be that vU1.3 isn't incorporated into U1 snRNPs, which seems antithetical to much of the thesis of the rest of the paper (that vU1.3 effects splicing via canonical interactions with the 5'SS) – I don't know whether this is simply because this type of RNA IP proteomics experiment is inherently extremely challenging, and/or whether it is the vU1.3 or the canonical U1 interactomes that are atypical, but I don't think it adds to (and to me detracts from) the rest of the manuscript.

Fig 2A needs to be clarified, as I'm confused what it shows – as written it says 'percent of all active genes', but 50% of active (expressed, so thousands of) genes can't have differential isoforms given the N=92/etc numbers in the results section; it seems like the x-axis of Fig. 2A is instead showing the percent of genes with differential splicing? Similarly I'm confused by the dotted line at 50% in Fig. 2A – it seems like the baseline of 'what percent of genes have ORF loss/gain' (etc) shouldn't all be 50%, but would change based on how frequent that property is genome-wide? Unless I'm completely misinterpreting the figure

I'm missing Methods text for the ENCODE analysis (Fig. S4A) – the only description is in the figure legend as 'mean eCLIP signal enrichment', but I'm unclear what that means – the mean of signal across all RS sites? The mean of fold-enrichment in IP versus control? I'm surprised by these findings as multiple of the listed RBPs (IGF2BP1/3, DDX6) are more canonically cytoplasmic proteins, which raises concern that the observations in Fig. S4A are coming from a small number of mis-annotated sites. More broadly, though, I'm not really clear how Fig. S4A, C, D connect to the rest of the manuscript - they

don't really seem to be connected to the vU1-dependent effects in the paper.

It seems that some figures were rearranged and not properly edited in the text – the final section of the results section refers throughout to Fig 4

Fig. 3i (in the figures, I believe in the text it's called out as Fig. 3j) seems to me the most critical panel in the latter story, but I think it needs some clarification – what is the Fisher's Exact test testing (lost vs unchanged)? It seems based superficially looking at the values that some other changes are larger than the two shown as significant, I'm surprised those do not also meet the P-value cutoff?

Minor comments:

I don't have an issue with the overall point (as it does seem that broadly vU1.3 & vU1.8 are both higher in embryonic tissues from S1a), but a Pearson correlation seems not correct to use with the degree of outliers in S1B; this point would probably more accurately be made by simply doing some sort of test (t-test / Kolmogorov-Smirnov test) on the vU1/U1 ratios from S1A on embryo vs non-embryo

I'm confused by the combination of Fig. 1I and Fig. S2J – the text implies that they're to be compared against each other (that Fig. 1I shows increased induction of differentiation markers with vU1.8 KO and Fig. S2J shows 'such effects could not be observed in vU1.3-KO cells', but 3 of the 4 markers are different between Fig. 1I and Fig. S2J ? I'm not sure how these are comparable if they're not showing the same markers. Fig 1I also has a typo "differentia[[n]]tion markers"

Fig 1K should have some statistical tests on the distributions to support text claims that the genes are suppressed or up-regulated.

I believe Fig. 2G is only referred in the text after Fig. 2H & I (also the Fig. 2i callout is typed on pg. 8 as Fig. 3i)

In the discussion, "First, and despite the fact that no discernible phenotypic changes were seen, the loss of vU1.8 results in cell cycle effects..." – I'd reword this, as the second part of the sentence discusses phenotypic changes that were discerned (perhaps "morphological" rather than "phenotypic"?)

"Our data suggest that vU1 binding to these RS sites can signal the skipping of such RS-exons, providing another layer of RNA quality control" – I would rephrase this, as I don't think the authors show vU1 binding to these exons?

Reviewer #2

(Remarks to the Author)

The manuscript by Papantonis and colleagues, entitled "Variant U1 snRNAs contribute to cell cycle and differentiation control of human iPS cells", explores the roles of two U1 snRNA variants, vU1.3 and vU1.8, which are highly expressed in embryonic tissues and organs. The authors provide an appealing rationale for studying these variants, as both diverge from canonical U1 in key features, suggesting distinct splicing activities. Notably, vU1.3 carries a C-to-T mutation in the 5' splice site, enabling recognition of AT instead of the canonical GT splice donor site, and may therefore contribute to recursive splicing. Both variants also contain base substitutions within the U1-70K binding domain. While the work is timely and addresses an important aspect of splicing regulation in hiPSCs, several claims, especially those concerning cell cycle control, require stronger experimental support.

Specific comments:

Fig.1

- Cell cycle claims

The title emphasizes cell cycle regulation, yet the manuscript lacks direct functional data supporting this connection. The authors rely mainly on differential expression and GO-term enrichment, with only a single bar plot (Fig. 1F) to illustrate cell cycle involvement.

To substantiate the link, considering that both main title and Figure 1 title refer to cell cycle, I would strongly expect to see a conventional population-based DNA content analysis by propidium iodide staining and FACS for each clone separately, or single-cell confocal fluorescence imaging for Cyclin B1 as a robust indicator of G2/M transition. Inclusion of drug treatment control (e.g. nocodazole, thymidine block) would provide essential benchmarks and allow comparison of vU1.8 KO cells with known cell cycle perturbations.

- RNA immunoprecipitation (RIP) experiment:

The RIP experiment lacks an adequate control. Using KO cells as background is not sufficient to confirm specificity. Scrambled RNA controls (same cellular background with irrelevant probes) should be included, analogous to IgG-only or tag-only controls in antibody-based assays. Without these, the interpretation of enrichment remains questionable.

- Comparison with differentiation datasets

The authors report that vU1.8-KO downregulated genes are enriched for G2/M terms, and that cells accumulate in G2. However, they also compare these findings to late-stage neurogenesis, which is typically characterized by cell cycle exit or G1 phase lengthening, not G2 accumulation. The biological rationale for this comparison should be clarified, or the analysis reframed.

Fig. 2

Consistent with U1's known role in suppressing premature cleavage/polyadenylation ("telescripting"), the majority of observed effects involved alternative transcription start (TS) and 3' end (TE) usage. Only a small overlap of isoform switches (6 genes) was shared between vU1.3 and vU1.8 KO, but these were enriched for key pathways (embryonic development, morphogenesis, p53 signaling).

The link between vU1 ablation (especially vU1.8-KO) and cell cycle regulation is pointing to GO enrichments (cell cycle, DNA metabolism, chromatin organization) and to a link between direct AS targets as cell cycle genes (ARPP19 as G2/M transition factors, CPSF7 as a pre-mRNA cleavage and polyadenylation factor that could link transcription termination with cell cycle regulation at G2/M checkpoint, while CTNNB1 and CCDC57 could be involved in G1/S checkpoint).

Albeit mechanistically plausible, the link between vU1.8 function and cell cycle regulation should be validated, as it remains solely associative

The proposed mechanistic link (e.g., AS of ARPP19, CPSF7, CCDC57, CTNNB1) is plausible, but requires validation, similar to what suggested above:

- Phenotype level: cell cycle profiling (flow cytometry, microscopy).
- Mechanism level: RT-PCR/qPCR validation of the highlighted AS events.
- Protein level: Western blot or targeted proteomics for key regulators.
- Causality: genetic rescue of vU1.8 expression or splice-switching approaches for target genes.

Figure 3 / Recursive splicing (RS):

- The observation that RS events are lost more frequently than gained in vU1.8-KO cells is intriguing, and their enrichment in developmental/differentiation pathways is consistent with the broader transcriptomic phenotype.
- However, there is no demonstrated correlation between RS events and the cell cycle regulators discussed earlier (ARPP19, CPSF7, etc.). While it is plausible that RS contributes indirectly to cell cycle gene regulation, this connection is not substantiated in the current dataset.
- At present, the RS analysis is descriptive and preliminary; stronger integration with functional assays would be needed to argue for a meaningful contribution to cell cycle control.

Overall Assessment

This study highlights a potentially important role of variant U1 snRNAs in shaping splicing programs in hiPSCs. However, the central claim regarding cell cycle regulation is insufficiently supported by functional evidence. The data convincingly demonstrate transcriptomic changes, including AS and RS events, but without complementary phenotypic assays, protein validation, or rescue experiments, the manuscript remains at an observational stage. Also, there is not a clear immediate link between RS and cell cycle progression, so a model would be greatly appreciated. Addressing these points would substantially strengthen the work and bring it in line with the claims in the title.

Reviewer #3

(Remarks to the Author)

This is a very interesting and well-written manuscript aimed at understanding the role of two variant forms of U1 snRNAs in human IPS cells. This is an important effort as there are many copies of each snRNA in the genome, many of which vary in sequence from the canonical snRNA sequence, and this has led to questions on whether these variant snRNAs have functional roles in splicing or not. The evaluation and manipulation of endogenous snRNA genes has been challenging given the large number of snRNAs in the genome and their homologous sequences. The authors study two specific variant U1 snRNAs here (vU1.3 and vU1.8) by CRISPR deletion and then study the impact on gene expression, splicing, and their RNA interactions in IPS cells. They find that these two variant U1 snRNAs indeed have a functional impact on splicing.

While the study is overall well done, there are several very important points that need to be addressed as follows:

- Given the homology in sequence of not only the U1 snRNAs but also their surrounding genomic sequences, even using CRISPR sgRNAs to modify U1 snRNA genes selectively has been challenging. More detailed information on how the specificity of their CRISPR KO approach selectively affects only vU1.3 and vU1.8 needs to be evaluated. The question is how is it known that >1 U1 snRNA gene has not been affected by the CRISPR sgRNA approach?
- The data used to identify vU1.3 and vU1.8 expression across human tissues needs more detail. Conventional bulk short-read RNA-seq data cannot resolve expression of individual U1 snRNA sequences (which has been a challenge in this area of study). Was long-read RNA-seq data used?
- Are vU1.3 and vU1.8 incorporated into U1 snRNP?
- The splicing analyses are interesting and well done but a presentation of the 5' splice site motifs enriched upon KO of vU1.3 and vU1.8 or the RNAs bound to them relative to canonical U1 snRNA would be important to present.

Reviewer #4

(Remarks to the Author)

Overall, the awkward writing and incorrect referencing of figures made this manuscript very difficult to read and follow. A careful read through and editing to correct grammatical errors and figure references is recommended. Authors occasionally make statements without including data to substantiate (see below). Abbreviations should be defined once and then used throughout the article (example: CE, RSS, AMT, etc.).

Major and minor comments are below.

- Page 3, paragraph 3: It is stated that “Despite the apparent differences in the magnitude and type of gene expression changes triggered, the two vU1 knockouts converge in that they both deregulate genes involved in nucleosome assembly, RNA metabolism and splicing, as well as in Wnt and Notch signaling (see Table S2) that are important for hiPSC differentiation.” From this sentence or Table S2 it is not clear how many and which genes pertaining to these processes are deregulated.
- Page 3, paragraph 4: The use of word “immunoprecipitation” is incorrect for RNA’s that are pulled down with the snRNAs.
- Page 4, legend to Figure 1: Use of “Molecular weight marker” for basepairs should be corrected.
- Page 4, paragraph 2: It is stated that “For vU1.3, we also managed to retrieve a protein interactome using the same IP approach.”; however, it is unclear what experiment was performed and how was the data analyzed.
- Page 6, the paragraph beginning with “Last, since the knockout of vU1.8...” makes claims based on Fig 1h-1K, but that significance values are not indicated for t-testing for the bar graphs, bar graphs, or box plots. Are there significant differences observed in the expression levels of the differentiation markers after vU1.8 KO?
- Page 7: What are the 6 isoform switch events shared between the two knockouts? Authors also mention “how enrichment for genes in important pathways like embryonic development, tube morphogenesis or p53 signaling.” Gene names are not provided.
- Page 8: In the paragraph beginning with “As alternative TE events were the most prevalent...”, the authors incorrectly reference their figures.
 - o Figure 2E for 3’ end shortening
 - o Figure 2H GO terms
 - o Figure 2I 3’UTR usage enrichment in vU1.8 KO cells
- Page 8: The order of the Figure 2 discussion is confusing and jumps around a lot.
- Page 9: There is no figure 4. Figure 3 has been referred to as Figure 4, which makes it confusing to read.
- First line of page 3: “...to the canonical U1, vU1.3 carries a...”
- Figure S2B: A vast majority, but not all copies, of vU1.3 appear to be deleted. Consider changing language in the text to reflect this.
- Based on Figure 1F, accumulation of cells in G2 was at the expense of cells in G1, but not in S. Authors report at the expense of G1/S.
- Figure S2D: Differs from representation of GO in Figure 1D, which has analogous analyses. Recommend standardizing representation.
- Figure S2F: it is unclear what was immunoprecipitated in the control lines, if all copies of vU1.3 and vU1.8 were knocked out in the baseline controls. Additionally, it is not specified what method was used to standardize one “control” when two lines were used as control.
- Remove the word “that” in the last sentence of the first paragraph on page 5
- Figure 1i: Text states analyses were performed on days 1 and 2, but figure shows days 1 and 3.
- Figure 1k: In the text, authors state that genes downregulated in absence of vU1.8 overlap with those that are most downregulated later in neurogenesis, but do not provide any overlap or specific genes in the figure (or provide any tables specifying downregulated neurogenesis genes). The provided figure seems to only show that downregulated genes decrease in expression and upregulated genes increase in expression (somewhat) throughout differentiation.
- For the sentence “Of the hundreds of isoform switch events detected upon knockout of each vU1 snRNA, only 6 are shared between the two KOs – but show enrichment for genes in important pathways like embryonic development, tube morphogenesis or p53 signaling” authors may want to consider including a figure or table to substantiate this statement, or provide a list of the 6 genes.
- In the first full paragraph on page 8, Fig 3h should be changed to Fig 2h. Additionally, Fig3i in this paragraph needs to be changed to Fig2i. Consider rearranging the figure or labeling so Fig2g follows Fig2H to match the order the figures are discussed in the text. Additionally, the sentence “This exon is included >20% less in mRNA in vU1.3-, but not in vU1.3-KO hiPSCs (Fig2g),” should read “This exon is included >20% less in mRNA in vU1.3-, but not in vU1.8-KO hiPSCs (Fig2g).”
- Figure S4b: Authors use HEK293 cells, but do not provide reasoning for use of this line over hiPSCs.
 - o Additionally, it is unclear how the left panel validates RSS intermediates. It may be helpful to include sequences of primers used for PCR.
- In the middle paragraph on page 9, Fig4b, Fig 4c, Fig 4d, and Fig 4e need to be changed to Fig 3b, Fig 3c, Fig 3d, and Fig 3e
- Figure S4c: The figure includes a line for “head size,” but the authors do not discuss this in the text
- Figure S4d: The axes may be mislabeled
- Figure 3g: Authors state that more RSSs were lost than gained in both vU1s, but the figure does not appear to reflect that
 - o It seems that Figure 3g is supposed to be associated with the first sentence in the following paragraph (This showed that vU1.3 ablation may be equally correlated with alternative usage of transcription start and end sites, as well as with core exon usage, whereas vU1.8 loss produces a similar profile but with a strong enrichment for differential CE usage (Fig 3g)).
 - o Additionally, authors state that genes carrying gained or lost RSSs in vU1.8-KO cells were enriched for pathways linked to early development and cell differentiation, however they do not provide a figure to show this.
- Figure 3h: Similar to above, it appears that this figure is supposed to be associated with the last sentence in the previous paragraph (In total, we identified ~200 and >230 differentially used RS sites in vU1.3- and vU1.8-KO cells, respectively. In both knockout lines, more RSSs were lost than gained (Fig 3h (maybe also Fig3i)).”
- Figure 3i: See previous bullet; this figure also appears to be associated with the sentence “In the case of the vU1.3-KO, we recorded a significant decrease in the usage of RSSs with an AT donor with a concomitant increase in GN usage (Fig 3i).”
 - o Additionally associated with Conversely, in vU1.8-KO cells, we recorded a significant drop in GT RS-donor usage, but with a concomitant increase in GV ones (V=A/C/G; Fig 3i).
- Figure 3J: It appears that this figure is supposed to go with the sentence “In fact, when it comes to the location of these alternatively included/skipped CEs, we found that the gain or loss of an RS event predominantly affect exons located downstream of the RS site, directly implicating recursive splicing in mRNA composition outcomes (Fig 3j).”

- Figure 3k: Supposed to be associated with this sentence: In our hiPSC lines, the vast majority of RS sites (89% or more) were followed by an RS-exon (Fig 3k).
- In the first paragraph on page 12, authors state that the cryptic alternative exons were under the control of vU1.3 or vU1.8, referencing figure S3b, however, in the figure, it appears that CE is only upregulated in vU1.8.
- The wrong figure is referenced again: “Second, in the absence of vU1.8, key transcription factor genes like SNAI1 and OTX2 showed aberrant expression patterns upon non-directed differentiation (Fig 1i).”
 - o Again: This agrees with the broad and early changes in the expression of many genes known to be differentially regulated during a neuronal differentiation time course (Fig 1k), including genes in the critical Wnt and Notch pathways.

Version 1:

Reviewer comments:

Reviewer #1

(Remarks to the Author)

In general, the authors have addressed my concerns

One note on the edited manuscript -

“On the other hand, vU1.3/4 targets showed enrichment for the exclusion of cassette exons often marked by the presence of an AT donor site (Fig 2d,g) matching the fact that vU1.3/4 carry a substitution in their 5' ss domains theoretically enabling them to recognize AT donors (Extended Data Fig 1c)“ – this sentence is making a claim (that they are ‘often... AT donor site(s)'), but Fig 2d doesn't show anything about AT donor sites, and 2g shows a single example. If the authors want to make this claim here, the number of AT donor site-events should be listed in the text or shown in a figure.

Reviewer #2

(Remarks to the Author)

The authors have responded to my comments and clarified the questions I raised. I can see that that a thorough check and editing process has been carried out, particularly in response to the comments raised by reviewer 4. I believe that the manuscript is now acceptable for publication.

Reviewer #3

(Remarks to the Author)

The authors have addressed my prior questions and comments comprehensively.

Open Access This Peer Review File is licensed under a Creative Commons Attribution 4.0 International License, which permits use, sharing, adaptation, distribution and reproduction in any medium or format, as long as you give appropriate credit to the original author(s) and the source, provide a link to the Creative Commons license, and indicate if changes were

made.

Point-by-point response to reviewers' comments

Reviewer #1 (Remarks to the Author):

The manuscript by Zhu and Sofiadis et al. describe analysis of the impact of stem cell knockout of two variants of the U1 snRNA (U1-3 and U1-8). Using cell biology and genomics, they reveal interesting specific effects of the individual knockout of these two U1 variants that have significant expression in stem cells, and show data suggesting these variants play particularly interesting roles in stem cell biology and potentially control unique recursive splicing sites.

We were glad to see that the Reviewer thinks that our data “*reveal interesting specific effects*” of U1 variant knockouts “*suggesting these variants play particularly interesting roles in stem cell biology*”.

To me, the RNA IP experiment is confusingly described – it’s described as ‘using the knockout cells as baseline controls’, but I’m unclear what that means. Fig. 1F has the control as a 3rd experiment, but then I’m not clear what cells are used for that control – is it a vU1.3/vU1.8 double KO? Or a mixture of the two single-knockouts? I also don’t think the analysis here is well-described – the methods simply states “vU1 pulldown targets were quantified with NOIseq”, but I don’t see any sort of mention of statistical tests or even what the criteria is for defining an RNA as ‘interacting’ (the results just says ‘>1400 RNAs interacting with...’, with the genes simply listed in Table S3, so unless I’m missing something I can’t find any discussion of what cutoffs or statistics were used to identify those 1400).

The Reviewer is correct; our description provided little detail. We have now remedied this in both the main **Results** text (pg. 3) and in the **Methods** section (pg. 12) by explaining the following. First, all RNA-IP experiments were performed in triplicates, but the amount of RNA being pulled-down was so little that we had to merge all three replicates into one tube to have enough material even for a “high sensitivity” library prep (we tried repeatedly with single replicates and never got proper libraries for sequencing). This library prep though, attaches UMIs to the ends of all RNA molecules that allows us to control for PCR amplification artefacts. Following sequencing, enrichments over controls were computed using NOIseq v. 2.34.0 with default parameters. This package was specifically designed to calculate differential enrichments in datasets with single replicates. In our case, RNAs crosslinked to vU1.3 or vU1.8 and pulled-down were considered enriched once they met the 0.05 *P*-value cutoff set by NOIseq; this produced the list shown in **Extended Data Table 3**. Critically, these enrichments were doubly controlled. On the one hand, we performed IPs using scrambled oligos in WT cells to control for non-specific interactions with highly abundant RNAs, while on the other, we performed IPs with the vU1.3- or vU1.8-targeting oligos, but using the respective vU1-KO iPSC lines as input in order to control for non-specific interactions of the oligos (which were essentially zero, hence the lack of any corresponding data). We hope that this makes clear our strategy and bottlenecks for these RNA-IPs.

I would suggest that the next paragraph (“We crossed the lists of differentially regulated gene...”) be toned down – to me it’s unclear whether the data as described can be used to make statements like ‘only 10% of DEGs were direct targets’. Trivially, every intron-containing gene will at some level be bound by U1 in the cell (and likely most others, due to U1’s telescripting activity); but U1 isn’t profiled here to provide a baseline of how easy it is to detect these interactions, and with only 1 replicate each of vU1.3 and vU1.8, I’m not sure it’s possible to confidently determine to what degree this pulldown

has saturated discovery of vU1.3 and vU1.8-interacting RNAs, or whether the overlaps here reflect just the technical difficulty in pulling down and sequencing U1-interacting RNAs.

We fully agree with the Reviewer on this point. Indeed, it is difficult to assert that our RNA-IP saturated discovery of vU1.3-/vU1.8-interacting RNAs (especially given their low abundances and unstable nature). Therefore, we have now toned down these claims by referring to “putative vU1 targets” (see for example pg. 3-4) thereby making this caveat clear.

I think the RNA IP -> proteomics experiment described in Fig. S2H-I likely needs some degree of replication or validation before publication. The straightforward implication of S2I would be that vU1.3 isn't incorporated into U1 snRNPs, which seems antithetical to much of the thesis of the rest of the paper (that vU1.3 effects splicing via canonical interactions with the 5'SS) – I don't know whether this is simply because this type of RNA IP proteomics experiment is inherently extremely challenging, and/or whether it is the vU1.3 or the canonical U1 interactomes that are atypical, but I don't think it adds to (and to me detracts from) the rest of the manuscript.

Much like for the RNA-IPs, IP-MS experiments were indeed extremely challenging due to the amounts of material being pulled down each time. Again, here we had to combine multiple replicates into a single tube to perform mass-spec measurements, but the resulting interactomes were sparse. Given that vU1.8 readout never worked well (and are not presented), that the vU1.3 interactome indeed looks incomplete (as the Reviewer suspects) and that vU1s were shown to incorporate into functional snRNPs before (Mabin *et al.*, 2021), we decided to remove the IP-MS data from the manuscript.

Fig 2A needs to be clarified, as I'm confused what it shows – as written it says 'percent of all active genes'; but 50% of active (expressed, so thousands of) genes can't have differential isoforms given the N=92/etc numbers in the results section; it seems like the x-axis of Fig. 2A is instead showing the percent of genes with differential splicing? Similarly, I'm confused by the dotted line at 50% in Fig. 2A – it seems like the baseline of 'what percent of genes have ORF loss/gain' (etc) shouldn't all be 50%, but would change based on how frequent that property is genome-wide? Unless I'm completely misinterpreting the figure.

We apologise for this. The plot in **Fig 2a** is the standard outcome of *IsoformSwitchAnalyzeR* showing (in the x-axis) the fraction of genes that display the isoform change indicated (see y-axis) in respect to all genes that show this type of change between the two conditions tested. For example, in vU1.3-KO cells, ~60% of all genes that change in ORF length actually end up with shorter ORFs compared to control cells; and this is statistically significant (given a 95% confidence interval). We now spell this out in the revised legend if **Fig 2a**.

I'm missing Methods text for the ENCODE analysis (Fig. S4A) – the only description is in the figure legend as 'mean eCLIP signal enrichment', but I'm unclear what that means – the mean of signal across all RS sites? The mean of fold-enrichment in IP versus control? I'm surprised by these findings as multiple of the listed RBPs (IGF2BP1/3, DDX6) are more canonically cytoplasmic proteins, which raises concern that the observations in Fig. S4A are coming from a small number of mis-annotated sites. More broadly, though, I'm not really clear how Fig. S4A, C, D connect to the rest of the manuscript - they don't really seem to be connected to the vU1-dependent effects in the paper.

The Reviewer brings up two valid points here. First, in the original **Extended Data Fig 4a** we had simply plotted mean eCLIP signal in the 1 kbp around all RS sites we annotated. Of the many ENCODE eCLIP datasets, only the six shown in the figure showed marked enrichment (and two controls are shown in comparison). Of these 6 RBPs, DDX6 and IGF2BP1/3 are indeed known to have canonical cytoplasmic roles, but the COMPARTMENTS resource (<https://compartments.jensenlab.org/Search>; Binder *et al.*, 2014) that combines various experimental and text mining evidence, claims that all three RBPs also have nuclear localization. Nonetheless, these signal enrichments come from positions close to/at RS-exon junctions and might also represent binding to spliced mRNAs that include those RS-exons—we have no way of ruling this possibility out. Overall, our rationale was that if RS sites showed specific and strong (relative to input) enrichment for RBP binding, this would hint towards their functional relevance. But, since we do not follow up on these observations, we have now removed this analysis to not overcomplicated the manuscript.

Second, in the current **Extended Data Fig 4a,b** we use GARLIC (Nikolic *et al.*, 2017) to annotate and analyse disease-linked SNPs in extended RS sequences with the same purpose as above, namely that if RS sites showed specific and strong association with such SNPs in a tissue-specific manner, this would hint towards their functional relevance. We think that this is indeed the case here, and we would rather keep this analysis for readers to see (and perhaps follow up), although we acknowledge that it might be a bit tangential to the main focus of our study.

It seems that some figures were rearranged and not properly edited in the text – the final section of the results section refers throughout to Fig 4

We thank the Reviewer for noticing this; we have now corrected all misannotations of figure panels.

Fig. 3i (in the figures, I believe in the text it's called out as Fig. 3j) seems to me the most critical panel in the latter story, but I think it needs some clarification – what is the Fisher's Exact test testing (lost vs unchanged)? It seems based superficially looking at the values that some other changes are larger than the two shown as significant, I'm surprised those do not also meet the P-value cutoff?

The Reviewer is correct; here, we were testing for lost (or gained) v unchanged dinucleotides relevant to those presumed to be recognised by vU1.3 (AT) or vU1.8 (GT). However, for the sake of completion, we have now added asterisks denoting significance to all instances where a $P < 0.05$ cutoff is met.

Minor comments:

I don't have an issue with the overall point (as it does seem that broadly vU1.3 & vU1.8 are both higher in embryonic tissues from S1a), but a Pearson correlation seems not correct to use with the degree of outliers in S1B; this point would probably more accurately made by simply doing some sort of test (t-test / Kolmogorov-Smirnov test) on the vU1/U1 ratios from S1A on embryo vs non-embryo

We only calculated Pearson's correlation coefficients to demonstrate that vU1.3 and vU1.8 largely co-express. But, as the Reviewer suggested, t-tests for both vU1.3 and vU1.8 confirmed that they are expressed more in embryonic tissues ($P = 5.74e-05$ and $1.41e-02$, respectively). This information has now been added to the revised **Results** text (see pg. 2).

I'm confused by the combination of Fig. 1I and Fig. S2J – the text implies that they're to be compared against each other (that Fig. 1I shows increased induction of differentiation markers with vU1.8 KO and Fig. S2J shows 'such effects could not be observed in vU1.3-KO cells', but 3 of the 4 markers are

different between Fig. 1I and Fig. S2J ? I'm not sure how these are comparable if they're not showing the same markers. Fig 1I also has a typo "differentia[[n]]tion markers"

The Reviewer is correct, the way we had presented the data was confusing. In fact, we performed RT-qPCR on a set of 9 stem cell and differentiation markers, and chose to present 4 in main Fig 1i that showed some change upon vU1.8-KO. In the case of the vU1.3-KO, no marker showed any statistically significant change and we simply presented 4 as exemplary. However, we understand how this can be confusing, and are now showing the exact same 4 markers in **Extended Data Fig 2j**—we thank the Reviewer for noticing this.

Fig 1K should have some statistical tests on the distributions to support text claims that the genes are suppressed or up-regulated.

Wilcoxon-Mann-Whitney testing have now been added to the figure panel (now **Fig 1l**).

I believe Fig. 2G is only referred in the text after Fig. 2H & I (also the Fig. 2i callout is typed on pg. 8 as Fig. 3i)

We apologise for these mistakes, which we have now corrected.

In the discussion, "First, and despite the fact that no discernible phenotypic changes were seen, the loss of vU1.8 results in cell cycle effects..." – I'd reword this, as the second part of the sentence discusses phenotypic changes that were discerned (perhaps "morphological" rather than "phenotypic"?)

We thank the Reviewer for the suggestion; we now use the term "morphological" instead.

"Our data suggest that vU1 binding to these RS sites can signal the skipping of such RS-exons, providing another layer of RNA quality control" – I would rephrase this, as I don't think the authors show vU1 binding to these exons?

The Reviewer is correct, we have now changed this sentence to read "Our data suggest that these RS sites can signal the skipping of such RS-exons in the presence of specific vU1s, providing another layer of RNA quality control" (pg. 9).

Reviewer #2 (Remarks to the Author):

The manuscript by Papantonis and colleagues, entitled “Variant U1 snRNAs contribute to cell cycle and differentiation control of human iPS cells”, explores the roles of two U1 snRNA variants, vU1.3 and vU1.8, which are highly expressed in embryonic tissues and organs. The authors provide an appealing rationale for studying these variants, as both diverge from canonical U1 in key features, suggesting distinct splicing activities. Notably, vU1.3 carries a C-to-T mutation in the 5' splice site, enabling recognition of AT instead of the canonical GT splice donor site, and may therefore contribute to recursive splicing. Both variants also contain base substitutions within the U1-70K binding domain. While the work is timely and addresses an important aspect of splicing regulation in hiPSCs, several claims, especially those concerning cell cycle control, require stronger experimental support.

We are glad to see that the Reviewer finds our work provides “an appealing rationale for studying [U1] variants” and that it “is timely and addresses an important aspect of splicing regulation in hiPSCs”. Below, we explain how we revised the manuscript to remedy the concerns that the Reviewer raises.

Specific comments: Fig.1

• Cell cycle claims

The title emphasizes cell cycle regulation, yet the manuscript lacks direct functional data supporting this connection. The authors rely mainly on differential expression and GO-term enrichment, with only a single bar plot (Fig. 1F) to illustrate cell cycle involvement. To substantiate the link, considering that both main title and Figure 1 title refer to cell cycle, I would strongly expect to see a conventional population-based DNA content analysis by propidium iodide staining and FACS for each clone separately, or single-cell confocal fluorescence imaging for Cyclin B1 as a robust indicator of G2/M transition. Inclusion of drug treatment control (e.g nocodazole, thymidine block) would provide essential benchmarks and allow comparison of vU1.8 KO cells with known cell cycle perturbations.

The bar plots presented in current **Fig 1g** reflect the numbers of cell cycle-stratified hiPSCs in each genotype, but we are happy to present the conventional DNA content analysis via PI-staining and FACS (now added to **Extended Data Fig 2c** alongside a control of cells stuck at the G2/M checkpoint due to treatment with the CDK1 inhibitor RO3306). In addition, we performed Cyclin B1 immunostainings coupled to quantification of signal intensities per KO, which corroborated our FACS analysis (**Extended Data Fig 2d**)—we thank the Reviewer for both suggestions.

• RNA immunoprecipitation (RIP) experiment:

The RIP experiment lacks an adequate control. Using KO cells as background is not sufficient to confirm specificity. Scrambled RNA controls (same cellular background with irrelevant probes) should be included, analogous to IgG-only or tag-only controls in antibody-based assays. Without these, the interpretation of enrichment remains questionable.

We apologise for not making this clear (as this point was also raised by Reviewer #1). The vU1 RNA-IP enrichments calculated were actually doubly controlled. On the one hand, we performed IPs using scrambled oligos in order to control for non-specific interactions with highly abundant RNAs, while on the other, we performed IPs with the vU1.3- or vU1.8-targeting oligos but using the respective vU1-KO iPSC lines as input in WT cells to control for non-specific interactions of the targeting oligos (which were essentially zero, hence the lack of any corresponding sequencing data). Following sequencing, enrichments over controls were calculated using NO1seq v. 2.34.0 with default parameters, and RNAs

crosslinked to vU1.3 or vU1.8 and pulled-down were considered enriched once they met a 0.05 *P*-value cutoff; this produced the lists shown in **Extended Data Table 3**. We hope this clarifies our strategy for these RNA-IPs, which we now explain in both the **Results** (pg. 3) and the **Methods** sections (pg. 12).

- *Comparison with differentiation datasets*

The authors report that vU1.8-KO downregulated genes are enriched for G2/M terms, and that cells accumulate in G2. However, they also compare these findings to late-stage neurogenesis, which is typically characterized by cell cycle exit or G1 phase lengthening, not G2 accumulation. The biological rationale for this comparison should be clarified, or the analysis reframed.

We feel that the Reviewer links these two observations (i.e., KO-induced cell cycle changes and more general gene expression changes in hiPSCs) more than our analysis does. Indeed, the vU1-KO leads to discernible cell cycle effects (discussed above), but also to pronounced gene expression changes in hiPSCs. But we did not compare these changes to late-stage neurogenesis; rather we used short-term (2-day) undirected differentiations to find that the vU1.8-KO perturbed the levels of both stem cell and differentiation markers (**Fig 1j**), one of which was OTX2. We then used public time-resolved data from the differentiation of hiPSCs into brain organoids, where six consecutive developmental stages were compared to the changes the vU1.8-KO inflicted on hiPSCs (**Fig 1l**). Therefore, late-stage neurogenesis, where indeed cell cycle exit/G1 lengthening manifest, represents only a subset of the comparisons performed, and nowhere do we attempt to functionally link these particular stages to the cell cycle effects seen in KO hiPSCs (please see text on pg. 4).

Fig. 2

Consistent with U1's known role in suppressing premature cleavage/polyadenylation ("telescripting"), the majority of observed effects involved alternative transcription start (TS) and 3' end (TE) usage. Only a small overlap of isoform switches (6 genes) was shared between vU1.3 and vU1.8 KOs, but these were enriched for key pathways (embryonic development, morphogenesis, p53 signaling). The link between vU1 ablation (especially vU1.8-KO) and cell cycle regulation is pointing to GO enrichments (cell cycle, DNA metabolism, chromatin organization) and to a link between direct AS targets as cell cycle genes (ARPP19 as G2/M transition factors, CPSF7 as a pre-mRNA cleavage and polyadenylation factor that could link transcription termination with cell cycle regulation at G2/M checkpoint, while CTNNB1 and CCDC57 could be involved in G1/S checkpoint).

Albeit mechanistically plausible, the link between vU1.8 function and cell cycle regulation should be validated, as it remains solely associative. The proposed mechanistic link (e.g., AS of ARPP19, CPSF7, CCDC57, CTNNB1) is plausible, but requires validation, similar to what suggested above:

- *Phenotype level: cell cycle profiling (flow cytometry, microscopy).*
- *Mechanism level: RT-PCR/qPCR validation of the highlighted AS events.*
- *Protein level: Western blot or targeted proteomics for key regulators.*
- *Causality: genetic rescue of vU1.8 expression or splice-switching approaches for target genes.*

We appreciate the suggestions, and have now done the following. First, we have offered the FACS and microscopy profiling of these cell cycle changes (**Extended Data Fig 2c,d**) as also discussed above. Second, we have now added multiple RT-qPCR validations of different kinds of exemplary AS events (see **Fig 2g,j** and **Extended Data Fig 3c**). However, the precise genome editing for endogenous splice-switching tests are too complex (and costly) to carry out in hiPSCs, especially given the generally

modest magnitude of AS changes that we recorded. Nonetheless, we have now toned down the respective parts of the **Results** to reflect the lack of a full mechanistic link between our KO cells and the specific changes leading to cell cycle changes.

Figure 3 / Recursive splicing (RS):

The observation that RS events are lost more frequently than gained in vU1.8-KO cells is intriguing, and their enrichment in developmental/differentiation pathways is consistent with the broader transcriptomic phenotype. However, there is no demonstrated correlation between RS events and the cell cycle regulators discussed earlier (ARPP19, CPSF7, etc.). While it is plausible that RS contributes indirectly to cell cycle gene regulation, this connection is not substantiated in the current dataset. At present, the RS analysis is descriptive and preliminary; stronger integration with functional assays would be needed to argue for a meaningful contribution to cell cycle control.

This is probably a misunderstanding, as we do not claim that the potential regulation of RS events by vU1s (especially by the vU1.8-KO producing the most pronounced effects) are directly linked to the cell cycle changes we documented. Our motivation for looking into the vU1-RS relationship comes from our previous postulation that “non-canonical” RS junctions maybe recognized and processed via vU1-snRNPs (see Kelly *et al.*, 2015; Georgomanolis *et al.*, 2016). Therefore, we wanted to highlight yet another possible layer in splicing regulation, where vU1s may be involved. Nevertheless, the Reviewer’s remark prompted us to look into those genes that contain differentially-regulated RS sites upon vU1.8-KO, and we found them linked to such GO terms as “p53 regulation pathway”, “Rho GTPase regulation”, and “actin filament-based processes” (data now added to **Fig 3g**), and we indeed see differential usage of RS-exons in RNA-seq data (data now added to **Extended Data Fig 4e**). Still, we think that RS changes only represent an additional regulatory layer that operates in addition to the gene expression and AS changes documented (in **Figs 1-2**) to regulate iPSC functions, including cell cycle progression—we clarify our rationale on pg. 6-7 of the **Results**.

Overall Assessment

This study highlights a potentially important role of variant U1 snRNAs in shaping splicing programs in hiPSCs. However, the central claim regarding cell cycle regulation is insufficiently supported by functional evidence. The data convincingly demonstrate transcriptomic changes, including AS and RS events, but without complementary phenotypic assays, protein validation, or rescue experiments, the manuscript remains at an observational stage. Also, there is not a clear immediate link between RS and cell cycle progression, so a model would be greatly appreciated. Addressing these points would substantially strengthen the work and bring it in line with the claims in the title.

Again, we appreciate the feedback and the assessment that our study “*highlights a potentially important role of variant U1 snRNAs in shaping splicing programs in hiPSCs*”, and we hope that our new experiments and changes now bring the manuscript closer to the goal described by the Reviewer. Regarding the model, we now provide a concept of how vU1s underlie RS regulation in **Fig 3k**.

Reviewer #3 (Remarks to the Author):

This is a very interesting and well-written manuscript aimed at understanding the role of two variant forms of U1 snRNAs in human IPS cells. This is an important effort as there are many copies of each snRNA in the genome, many of which vary in sequence from the canonical snRNA sequence, and this has led to questions on whether these variant snRNAs have functional roles in splicing or not. The evaluation and manipulation of endogenous snRNA genes has been challenging given the large number of snRNAs in the genome and their homologous sequences. The authors study two specific variant U1 snRNAs here (vU1.3 and vU1.8) by CRISPR deletion and then study the impact on gene expression, splicing, and their RNA interactions in IPS cells. They find that these two variant U1 snRNAs indeed have a functional impact on splicing.

We thank the Reviewer for highlighting that our manuscript is “*very interesting and well-written*” and that it provides evidence on how “*variant U1 snRNAs indeed have a functional impact on splicing*”.

While the study is overall well done, there are several very important points that need to be addressed as follows:

-Given the homology in sequence of not only the U1 sRNAs but also their surrounding genomic sequences, even using CRISPR sgRNAs to modify U1 snRNA genes selectively has been challenging. More detailed information on how the specificity of their CRISPR KO approach selectively affects only vU1.3 and vU1.8 needs to be evaluated. The question is how is it known that >1 U1 snRNA gene has not been affected by the CRISPR sgRNA approach?

The vU1 genes we targeted were selected based on the combination of two criteria. First, that they be well expressed in hiPSCs; second, that variant-specific gRNAs could be designed without off-targets on other subfamily members. As these KOs initiated in my lab already in 2016 as part of the national Priority Program SPP1935 “*Deciphering the mRNP code*”, we used the existing nomenclature and annotation (according to O’Reilly *et al.*, 2012) to design the gRNAs shown in **Extended Data Table 1**. These appeared specific and we could verify a near-complete genome editing of hiPSCs in either case (**Fig 1b** and **Extended Data Fig 2b**). Still, motivated by the Reviewer’s comment about additional validations, we used primer pairs targeting vU1 genes via genomic DNA PCRs (**Extended Data Fig 1d** and **Table 6**) and saw that in the latest genome reference build (hg38) the vU1.3 and vU1.4 genes appear almost identical (despite not belonging to the same subgroup according to O’Reilly *et al.*, 2012). To this end, designing primers that cover both well, we found that copies of both are indeed deleted in hiPSCs (see **Figure 1 for Reviewers**, below). We therefore updated the text to reflect this new annotation (i.e. vU1.3 is now referred to as vU1.3/.4 throughout), while also adding a sequence alignment in **Extended Data Fig 1d**. This does not change interpretation of our results, as both vU1.3 and vU1.4 potentially recognize AT donors, while vU1.8 specificity remained unchallenged.

Reviewer Fig 1. Electrophoresis of gDNA PCR products testing for the specificity of CRISPR-mediated deletions of vU1.3 in hiPSCs. Historically, vU1.3 was grouped as similar to vU1.2a that is not affected by the gRNAs used for the deletion, while PCR of the near-identical vU1.3 and -1.4 genomic sequences reveal an essentially complete deletion of these loci. M: size marker profile.

-The data used to identify vU1.3 and vU1.8 expression across human tissues needs more detail. Conventional bulk short-read RNA-seq data cannot resolve expression of individual U1 snRNA sequences (which has been a challenge in this area of study). Was long-read RNA-seq data used?

Total RNA-seq data (which is what we used here) contain enough reads that can be uniquely mapped to many of these variants so as to provide a reasonable picture of the vU1 expression landscape (see **Figure 2 for Reviewers**, below). In fact, this rudimentary analysis is sensitive enough to identify the specific differential response of vU1s targeted for KO against the essentially unchanging background of expression of all other vU1s. Of course, many vU1s cannot be sufficiently mapped and long-read sequencing might remedy this, but we have little experience and access to this technology currently. In lieu of this, we have added RT-qPCR data showing specific vU1 expression changes in wt and KO hiPSCs (**Fig 1c** and **Extended Data Fig 2a**).

-Are vU1.3 and vU1.8 incorporated into U1 snRNP?

On top of our functional analyses, it has been reported that indeed they do (Mabin *et al.*, 2021) and we do refer to this previous work in both our **Introduction** and **Results** text.

-The splicing analyses are interesting and well done but a presentation of the 5' splice site motifs enriched upon KO of vU1.3 and vU1.8 or the RNAs bound to them relative to canonical U1 snRNA would be important to present.

We saw essentially no difference in 5' ss splice site motifs between wild-type and vU1-KO hiPSCs, hence the absence of any logos plotted (we now mention this on pg. 5 of the revised **Results**). The instance where we indeed saw selective motif changes were once recursive splicing was analysed, and these changes were reported in **Fig 3i,j** and discussed on pg. 8 of the revised **Results**.

Reviewer #4 (Remarks to the Author):

Overall, the awkward writing and incorrect referencing of figures made this manuscript very difficult to read and follow. A careful read through and editing to correct grammatical errors and figure references is recommended. Authors occasionally make statements without including data to substantiate (see below). Abbreviations should be defined once and then used throughout the article (example: CE, RSS, AMT, etc.).

The Reviewer is correct. We had inadvertently cited figures (especially latter ones in the manuscript) improperly, and we understand that this made the story difficult to follow. We apologise for this, and have now corrected everything, including grammatical errors and phrasing throughout the text.

Major and minor comments are below.

- *Page 3, paragraph 3: It is stated that “Despite the apparent differences in the magnitude and type of gene expression changes triggered, the two vU1 knockouts converge in that they both deregulate genes involved in nucleosome assembly, RNA metabolism and splicing, as well as in Wnt and Notch signaling (see Table S2) that are important for hiPSC differentiation.” From this sentence or Table S2 it is not clear how many and which genes pertaining to these processes are deregulated.*

The Reviewer is correct. We now provide some characteristic examples of genes associated with these GO terms in the **Results** text (e.g., *NOTCH1/3* or *SRSF2/3/6/7* on pg. 3). We also comment on the very limited overlap of similarly regulated genes between the two KOs (8 shared down- and only 1 shared upregulated gene) on the same page of the manuscript.

- *Page 3, paragraph 4: The use of word “immunoprecipitation” is incorrect for RNA’s that are pulled down with the snRNAs.*

This is correct. As biotinylated probes rather than antisera were used, we now use the term “co-purification” when referring to these pull-downs.

- *Page 4, legend to Figure 1: Use of “Molecular weight marker” for basepairs should be corrected.*

This has now been corrected to “size marker”.

- *Page 4, paragraph 2: It is stated that “For vU1.3, we also managed to retrieve a protein interactome using the same IP approach.”; however, it is unclear what experiment was performed and how was the data analyzed.*

We apologise for the vague statement. However, we have now opted to remove this data altogether due to the possibility that our IP was likely not saturated (see our response to Reviewer #1 on this).

- *Page 6, the paragraph beginning with “Last, since the knockout of vU1.8...” makes claims based on Fig 1h-1K, but that significance values are not indicated for t-testing for the bar graphs, bar graphs, or box plots. Are there significant differences observed in the expression levels of the differentiation markers after vU1.8 KO?*

The Reviewer is correct, significance testing was missing and we now added this (see new **Fig 1l**).

- *Page 7: What are the 6 isoform switch events shared between the two knockouts? Authors also mention “how enrichment for genes in important pathways like embryonic development, tube morphogenesis or p53 signaling.” Gene names are not provided.*

There are 6 types of events that shared between the two vU1-knockouts, and these concern only 15 genes that are also shared (all shown in new **Extended Data Fig 3b**). Of these we highlight *PDE4C*, *SORBS2*, *KLK8*, *PCDH11Y* and *STX1A* in the revised **Results** (on pg. 5) as they become induced during neuronal differentiation.

- *Page 8: In the paragraph beginning with “As alternative TE events were the most prevalent...”, the authors incorrectly reference their figures.*

- o *Figure 2E for 3’ end shortening*

- o *Figure 2H GO terms*

- o *Figure 2I 3’UTR usage enrichment in vU1.8 KO cells*

We apologise for this misreferencing, which we have now corrected.

- *Page 8: The order of the Figure 2 discussion is confusing and jumps around a lot.*

We have now reordered **Fig 2** panels to match the text’s flow. Thank you for noticing this.

- *Page 9: There is no figure 4. Figure 3 has been referred to as Figure 4, which makes it confusing to read.*

Apologies once again, we have corrected all references to figure panels.

- *First line of page 3: “...to the canonical U1, vU1.3 carries a...”*

We have now rewritten this part.

- *Figure S2B: a vast majority, but not all copies, of vU1.3 appear to be deleted. Consider changing language in the text to reflect this.*

This is correct; we have aligned the text to reflect this, while referring to vU1.3/.4 to match the most recent annotation for these genes (see top of pg. 3, **Extended Data Fig 1c** and **Figure 1 for Reviewers, above**).

- *Based on Figure 1F, accumulation of cells in G2 was at the expense of cells in G1, but not in S. Authors report at the expense of G1/S.*

Correct, the sentence now reads “...at the expense of cells in G1”. Thank you for pointing this out (which is also corroborated by the new FACS profiles and IFs in **Extended Data Fig 2c,d**).

- *Figure S2D: Differs from representation of GO in Figure 1D, which has analogous analyses. Recommend standardizing representation.*

We have now homogenized all representations of GO terms to match that in **Extended Data Fig 2f**.

- *Figure S2F: it is unclear what was immunoprecipitated in the control lines, if all copies of vU1.3 and vU1.8 were knocked out in the baseline controls. Additionally, it is not specified what method was used to standardize one “control” when two lines were used as control.*

We apologise for not making this clear (as this point was also raised by Reviewer #1). The vU1 RNA-IP enrichments calculated were actually doubly controlled. On the one hand, we performed IPs in WT cells using scrambled oligos to control for non-specific interactions with highly abundant RNAs, while on the other, we performed IPs with the vU1.3- or vU1.8-targeting oligos but using the respective vU1-

KO iPSC lines as input in order to control for non-specific interactions of the targeting oligos (which were essentially zero, hence the lack of corresponding sequencing data). Following sequencing, enrichments over controls were calculated using NOIseq v. 2.34.0 with default parameters, and RNAs crosslinked to vU1.3 or vU1.8 and pulled-down were considered enriched once they met a 0.05 *P*-value cutoff; this produced the two lists shown in **Extended Data Table 3**. We hope that this clarifies our strategy, which we now explain in both the **Results** (pg. 3) and the **Methods** sections (pg. 12)

- *Remove the word “that” in the last sentence of the first paragraph on page 5*

Now removed. Thank you for noticing.

- *Figure 1i: Text states analyses were performed on days 1 and 2, but figure shows days 1 and 3.*

The correct times are indeed days 1 and 2 of undirected differentiation; **Fig 1i** has been corrected.

- *Figure 1k: In the text, authors state that genes downregulated in absence of vU1.8 overlap with those that are most downregulated later in neurogenesis, but do not provide any overlap or specific genes in the figure (or provide any tables specifying downregulated neurogenesis genes). The provided figure seems to only show that downregulated genes decrease in expression and upregulated genes increase in expression (somewhat) throughout differentiation.*

This is a misunderstanding. For the genes behind each box plot in new **Fig 1l**, we simply crossed the vU1.8-KO DEGs with DEGs at the corresponding day of the iPSC-to-brain organoid differentiation (data from Zafeiriou et al., 2020). Our intention, which we now clarify in the **Results** text, was to test whether genes that are supposed to change during development (both early and late) are already affected upon vU1-KOs and to which extent. This proved true (see modified **Fig 1l** that shows log₂FC values for more intuitive interpretation), especially for downregulated genes. For a glimpse at the actual genes behind this simple comparison, the Reviewer can visit the following online file:

https://wwwuser.gwdguser.de/~yzhu1/0vF36w1M7UjV/RS/Analysis/5_Metascrape/neuron_related_DEGs_vU1KO_yajie.xlsx

- *For the sentence “Of the hundreds of isoform switch events detected upon knockout of each vU1 snRNA, only 6 are shared between the two KOs – but show enrichment for genes in important pathways like embryonic development, tube morphogenesis or p53 signaling” authors may want to consider including a figure or table to substantiate this statement, or provide a list of the 6 genes.*

As mentioned in response to a similar comment above, these genes are now listed in **Extended Data Fig 3b**, while we refer to them only once in the main **Results** text (see pg. 5).

- *In the first full paragraph on page 8, Fig 3h should be changed to Fig 2h. Additionally, Fig3i in this paragraph needs to be changed to Fig2i. Consider rearranging the figure or labeling so Fig2g follows Fig2H to match the order the figures are discussed in the text. Additionally, the sentence “This exon is included >20% less in mRNA in vU1.3-, but not in vU1.3-KO hiPSCs (Fig2g),” should read “This exon is included >20% less in mRNA in vU1.3-, but not in vU1.8-KO hiPSCs (Fig2g).”*

The Reviewer is again correct; we have now implemented all these changes to the text.

- *Figure S4b: Authors use HEK293 cells, but do not provide reasoning for use of this line over hiPSCs.*

o Additionally, it is unclear how the left panel validates RSS intermediates. It may be helpful to include sequences of primers used for PCR.

HEK293 cells were used here for three reasons: they are significantly cheaper to grow, significantly easier to handle when selecting single edited clones, and exemplify that recursive splicing can be seen as a universal mechanism extending to non-hiPS cells. The left-hand-side panel of **Extended Data Fig 4a** simply reflects PCR on genomic DNA following biallelic deletion of two different RS sites, which were validated by Sanger sequencing and then tested via RT-qPCR for effects on splicing of exons flanking the deleted site. These details have now been clarified in the revised text (pg. 7). The oligos used as gRNAs or for gDNA PCR have been included in **Extended Data Tables 1** and **6**.

• In the middle paragraph on page 9, Fig4b, Fig 4c, Fig 4d, and Fig 4e need to be changed to Fig 3b, Fig 3c, Fig 3d, and Fig 3e

The Reviewer is once again correct, and we have now made these changes.

• Figure S4c: The figure includes a line for “head size,” but the authors do not discuss this in the text

This is a trait that should be unrelated to heart development and its cell types, and serves as a control. We explain this in the respective figure legend (currently **Extended Data Fig 4b**).

• Figure S4d: The axes may be mislabeled

Axis labels are correct, but the zoom-ins were rotated with respect to the main heatmap; we have now corrected this to make interpretation by the readers more intuitive.

• Figure 3g: Authors state that more RSSs were lost than gained in both vU1s, but the figure does not appear to reflect that.

o It seems that Figure 3g is supposed to be associated with the first sentence in the following paragraph (This showed that vU1.3 ablation may be equally correlated with alternative usage of transcription start and end sites, as well as with core exon usage, whereas vU1.8 loss produces a similar profile but with a strong enrichment for differential CE usage (Fig 3g)).

o Additionally, authors state that genes carrying gained or lost RSSs in vU1.8-KO cells were enriched for pathways linked to early development and cell differentiation, however they do not provide a figure to show this.

The correct reference for gained/lost RSSs is actually **Fig 3i**, and the relevant GO terms/pathways are now shown in **Fig 3g**.

• Figure 3h: Similar to above, it appears that this figure is supposed to be associated with the last sentence in the previous paragraph (In total, we identified ~200 and >230 differentially used RS sites in vU1.3- and vU1.8-KO cells, respectively. In both knockout lines, more RSSs were lost than gained (Fig 3h (maybe also Fig3i)).”

• Figure 3i: See previous bullet; this figure also appears to be associated with the sentence “In the case of the vU1.3-KO, we recorded a significant decrease in the usage of RSSs with an AT donor with a concomitant increase in GN usage (Fig 3i).

o Additionally associated with Conversely, in vU1.8-KO cells, we recorded a significant drop in GT RS-donor usage, but with a concomitant increase in GV ones (V=A/C/G; Fig 3i).

• *Figure 3j: It appears that this figure is supposed to go with the sentence “In fact, when it comes to the location of these alternatively included/skipped CEs, we found that the gain or loss of an RS event predominantly affect exons located downstream of the RS site, directly implicating recursive splicing in mRNA composition outcomes (Fig 3j).”*

• *Figure 3k: Supposed to be associated with this sentence: In our hiPSC lines, the vast majority of RS sites (89% or more) were followed by an RS-exon (Fig 3k).*

All the aforementioned panels of **Fig 3** have now been renumbered and assigned to the appropriate place in the revised **Results** section; we thank the Reviewer for pointing all these mistakes out, and apologise once again for the mix up that certainly made the text more difficult than it should be.

• *In the first paragraph on page 12, authors state that the cryptic alternative exons were under the control of vU1.3 or vU1.8, referencing figure S3b, however, in the figure, it appears that CE is only upregulated in vU1.8.*

Again, this is a mis-annotation, which has now been corrected (also by the addition of new data like that in **Extended Data Fig 4e** about potentially cryptic RS-exons).

• *The wrong figure is referenced again: “Second, in the absence of vU1.8, key transcription factor genes like SNAI1 and OTX2 showed aberrant expression patterns upon non-directed differentiation (Fig 1i).”*

o Again: This agrees with the broad and early changes in the expression of many genes known to be differentially regulated during a neuronal differentiation time course (Fig 1k), including genes in the critical Wnt and Notch pathways.

Thank you once again for pointing out this mistake, which has now also been corrected.

Point-by-point response to reviewers' comments

Reviewer #1 (Remarks to the Author):

In general, the authors have addressed my concerns. One note on the edited manuscript – “On the other hand, vU1.3/.4 targets showed enrichment for the exclusion of cassette exons often marked by the presence of an AT donor site (Fig 2d,g) matching the fact that vU1.3/.4 carry a substitution in their 5' ss domains theoretically enabling them to recognize AT donors (Extended Data Fig 1c)” – this sentence is making a claim (that they are ‘often... AT donor site(s)'), but Fig 2d doesn't show anything about AT donor sites, and 2g shows a single example. If the authors want to make this claim here, the number of AT donor site-events should be listed in the text or shown in a figure.

We understand the Reviewer's point and have removed this qualifying statement from the main text (see pg. 6). We are also happy to see that all concerns have now been addressed.

Reviewer #2 (Remarks to the Author):

The authors have responded to my comments and clarified the questions I raised. I can see that that a thorough check and editing process has been carried out, particularly in response to the comments raised by reviewer 4. I believe that the manuscript is now acceptable for publication.

We thank the Reviewer for this endorsement for publication.

Reviewer #3 (Remarks to the Author):

The authors have addressed my prior questions and comments comprehensively.

We thank the Reviewer for this acknowledgement.